# Choroid plexus NKCC1 mediates cerebrospinal fluid clearance during mouse early postnatal development

Huixin Xu [1,12], Ryann M. Fame [1,12], Cameron Sadegh [1,2], Jason Sutin[3], Christopher Naranjo[4], Della Syau[4], Jin Cui[1], Frederick B. Shipley[1,5], Amanda Vernon[6,7,8], Fan Gao[6,7,8,11], Yong Zhang[9], Michael J. Holtzman[9], Myriam Heiman[6,7,8], Benjamin C. Warf[10], Pei-Yi Lin[3] & Maria K. Lehtinen [1,5 ✉]

Cerebrospinal fluid (CSF) provides vital support for the brain. Abnormal CSF accumulation, such as hydrocephalus, can negatively affect perinatal neurodevelopment. The mechanisms regulating CSF clearance during the postnatal critical period are unclear. Here, we show that CSF $K^+$, accompanied by water, is cleared through the choroid plexus (ChP) during mouse early postnatal development. We report that, at this developmental stage, the ChP showed increased ATP production and increased expression of ATP-dependent $K^+$ transporters, particularly the $Na^+$, $K^+$, $Cl^-$, and water cotransporter NKCC1. Overexpression of NKCC1 in the ChP resulted in increased CSF $K^+$ clearance, increased cerebral compliance, and reduced circulating CSF in the brain without changes in intracranial pressure in mice. Moreover, ChP-specific NKCC1 overexpression in an obstructive hydrocephalus mouse model resulted in reduced ventriculomegaly. Collectively, our results implicate NKCC1 in regulating CSF $K^+$ clearance through the ChP in the critical period during postnatal neurodevelopment in mice.

[1] Department of Pathology, Boston Children's Hospital, Boston, MA 02115, USA. [2] Department of Neurosurgery, Massachusetts General Hospital and Harvard Medical School, Boston, MA 02114, USA. [3] Fetal-Neonatal Neuroimaging and Developmental Science Center, Division of Newborn Medicine, Boston Children's Hospital, Harvard Medical School, 300 Longwood Avenue, Boston, MA 02115, USA. [4] Summer Honors Undergraduate Research Program, Division of Medical Sciences, Harvard Medical School, Boston, MA 02115, USA. [5] Graduate Program in Biophysics, Harvard University, Cambridge, MA 02138, USA. [6] Broad Institute of MIT and Harvard, Cambridge, MA 02142, USA. [7] Picower Institute for Learning and Memory, Cambridge, MA 02139, USA. [8] Department of Brain and Cognitive Sciences, Massachusetts Institute of Technology, Cambridge, MA 02139, USA. [9] Pulmonary and Critical Care Medicine, Department of Medicine, Washington University, St. Louis, MO 63110, USA. [10] Department of Neurosurgery, Boston Children's Hospital, Boston, MA 02115, USA. [11]Present address: Bioinformatics Resource Center in the Beckman Institute at Caltech, Pasadena, CA 91125, USA. [12]These authors contributed equally: Huixin Xu, Ryann M Fame. ✉email: maria.lehtinen@childrens.harvard.edu

A balance between cerebrospinal fluid (CSF) production and clearance (influx/efflux) is essential for normal brain function and development[1]. Disrupted CSF volume homeostasis with excessive CSF accumulation is implicated in pediatric brain disorders, in particular congenital hydrocephalus[2], where patients suffer from a potentially life-threatening accumulation of CSF and frequently develop neurological deficits that last through childhood and into adult life[3]. Often schizophrenia patients have enlarged lateral ventricles by their first episode of psychosis[4], in some cases as early as infancy[5], suggesting a role for CSF clearance abnormalities in this and possibly other neurodevelopmental disorders. As another example, autism spectrum disorders are associated with altered CSF distribution patterns and enlarged CSF space surrounding the brain[6]. A better understanding of developing CSF dynamics may help explain why early phases of brain development (e.g., from third trimester to 6 months after birth in human) represent a period of high vulnerability to certain congenital disorders[5–7].

Critically, how CSF is cleared during this perinatal period remains a mystery. Progress in CSF dynamics research has identified several putative CSF clearance routes including arachnoid villi and granulations in human (and only arachnoid villi in non-human mammalian species), perineural and paravascular pathways, and meningeal lymphatics[1,8,9]. While many mechanistic questions and validation work remain regarding the identified routes[8–10], their developmental time-courses are even less understood. In human, arachnoid granulations are not fully formed until 2 years of age. In other mammalian species (rats, pigs, and sheep), lymphatic vessels contribute to CSF clearance by connecting with CSF spaces around the cribriform plate and olfactory nerve roots during perinatal brain development[11–13]. Similarly, lymphatic structures begin to sprout at the base of the skull around birth in mice, but mature throughout the first postnatal month[14]. Thus, available data suggest that the above-mentioned CSF clearance routes are unlikely to fully account for CSF outflow during the critical and sensitive early phase of brain development. Identifying and manipulating the early endogenous CSF clearance mechanisms could provide one powerful approach for tackling neurodevelopmental disorders involving CSF dysregulation, and may also be applied to fluid disorders affecting adults.

To identify how early CSF is cleared, we investigated tissues with the ability to modulate CSF at this stage. The choroid plexus (ChP) is an intraventricular epithelial structure that forms the majority of the blood-CSF barrier and emerges prenatally. It contains diverse ion and fluid transporters along its vast surface area[15]. Although the prevailing model posits that the ChP provides net luminal secretion of ions and water to form CSF in adult brains[16–18], historical clinical observations suggest some absorptive functions of the ChP[19], which is supported by animal studies[20]. Moreover, extensive studies have documented ion gradients driving bidirectional trafficking at the ChP-CSF interface through various ChP transporters[15,21,22]. Furthermore, broad transcriptional changes of the machinery regulating fluid/ion transport support the concept of temporally dynamic and possibly context-dependent ChP functions in determining net directionality of CSF transport[23,24].

To further explore potentially absorptive properties of the early ChP, we studied the expression of transporters, the energy systems and ionic gradients that govern their activity, and their physiological effects across the timespan of early postnatal development in mice. Taken together our data support a previously undescribed developmental mechanism for net CSF clearance by the $Na^+$-$K^+$-$Cl^-$ and water cotransporter, NKCC1, on the apical membrane of the ChP during a specific stage. While NKCC1 retains bidirectional transporter potential throughout

life, its role in mediating net CSF-to-ChP transport during the early postnatal period of brain development critically influenced the establishment of CSF ion and fluid homeostasis. These results have implications for the pathophysiology of congenital disorders accompanied by dysregulated CSF and could inform strategies for treatment of neonatal hydrocephalus and perhaps other disorders.

## Results

**CSF $K^+$ declines precipitously during a specific perinatal period.** We discovered a unique and transient phase of neurodevelopment when CSF $[K^+]$ decreased rapidly. We used inductively coupled plasma–optical emission spectrometry (ICP-OES) and ion chromatography (IC) to measure levels of key ions likely to govern CSF flux including $Na^+$, $K^+$, and $Cl^-$ at several developmental timepoints. CSF $[K^+]$ was remarkably high at birth (9.6 ± 3.5 mM), decreased rapidly to 4.4 ± 0.9 mM by P7 (Fig. 1a), and later achieved adult levels of 3.1 ± 0.6 mM (Fig. 1a) while $[Na^+]$ was minimally changed and $[Cl^-]$ slightly increased (Supplementary Table 1 shows developmental CSF $[K^+]$, $[Na^+]$, $[Ca^{2+}]$, $[Mg^{2+}]$, $[Cl^-]$). Notably, the CSF-to-serum $[K^+]$ ratio decreased sharply from P0 to P7 (Fig. 1b), suggesting that the onset of CSF clearance occurs during this window[25]. Similar trends of CSF $[K^+]$ decrease as development proceeds have been reported in many other species perinatally, although by far none have shown a decrease as drastic as in mice, perhaps reflecting different stages of brain development at the times of sampling[17,26]. Studies in adult animals have shown rapid clearance of $K^+$ out of the CSF when the concentrations are higher than homeostatic levels[27–29], supporting the concept that CSF $[K^+]$-sensitive mechanisms exist and can mediate $K^+$ clearance in a timely manner. Notably, $K^+$ transport has been associated with water co-transport by several $K^+$ transporters in various tissues and cell types[30–32], suggesting that CSF $[K^+]$ changes could drive water movement in the brain as well. Therefore, we sought to identify mechanisms underlying this fast clearance of CSF $K^+$, which may shed light on CSF outflow during this time.

**ChP metabolic rate increases during the early postnatal transition phase.** The transitional period of rapid CSF $K^+$ clearance coincided with high ChP metabolism (Fig. 1a, b, h, i). We reasoned that $K^+$ clearance during this period could be ChP-mediated because the ChP expresses high levels of $K^+$ cotransporters on its large CSF-contacting surface area[15,33]. Similar to water and ion transport by other epithelial structures such as kidney proximal and distal tubes[34], $K^+$ clearance from CSF by the ChP would be energy-dependent and therefore be accompanied by upregulation of ATP production and mitochondrial activity. Therefore, we evaluated the metabolic status and ATP production capacity of the ChP epithelium before, during, and after the time period of CSF $[K^+]$ reduction. We found that both mitochondria number and size increased from E16.5 to 2 months old (2mo, referred to as "adult") (Fig. 1c–f), while cellular glycogen load gradually decreased (Supplementary Fig. 1). Both observations are consistent with reports from ChP in other mammalian species[33,35] and suggest functional changes in ChP oxidative metabolism. We used Agilent Seahorse XFe technology to monitor oxygen consumption as an index of the metabolic status of the ChP in explants from embryonic day 16.5 (E16.5), postnatal day 0 (P0), P7, and adult mice. We then calculated basal metabolism and ATP production (Fig. 1g–i, Supplementary Fig. 2). E16.5 ChP had the lowest basal respiration of all tested ages (Fig. 1h, Supplementary Fig. 2). Adult had a higher capacity for overall ATP production than E16.5 ChP, but surprisingly, P0–P7 ChP were the most metabolically active as per ATP

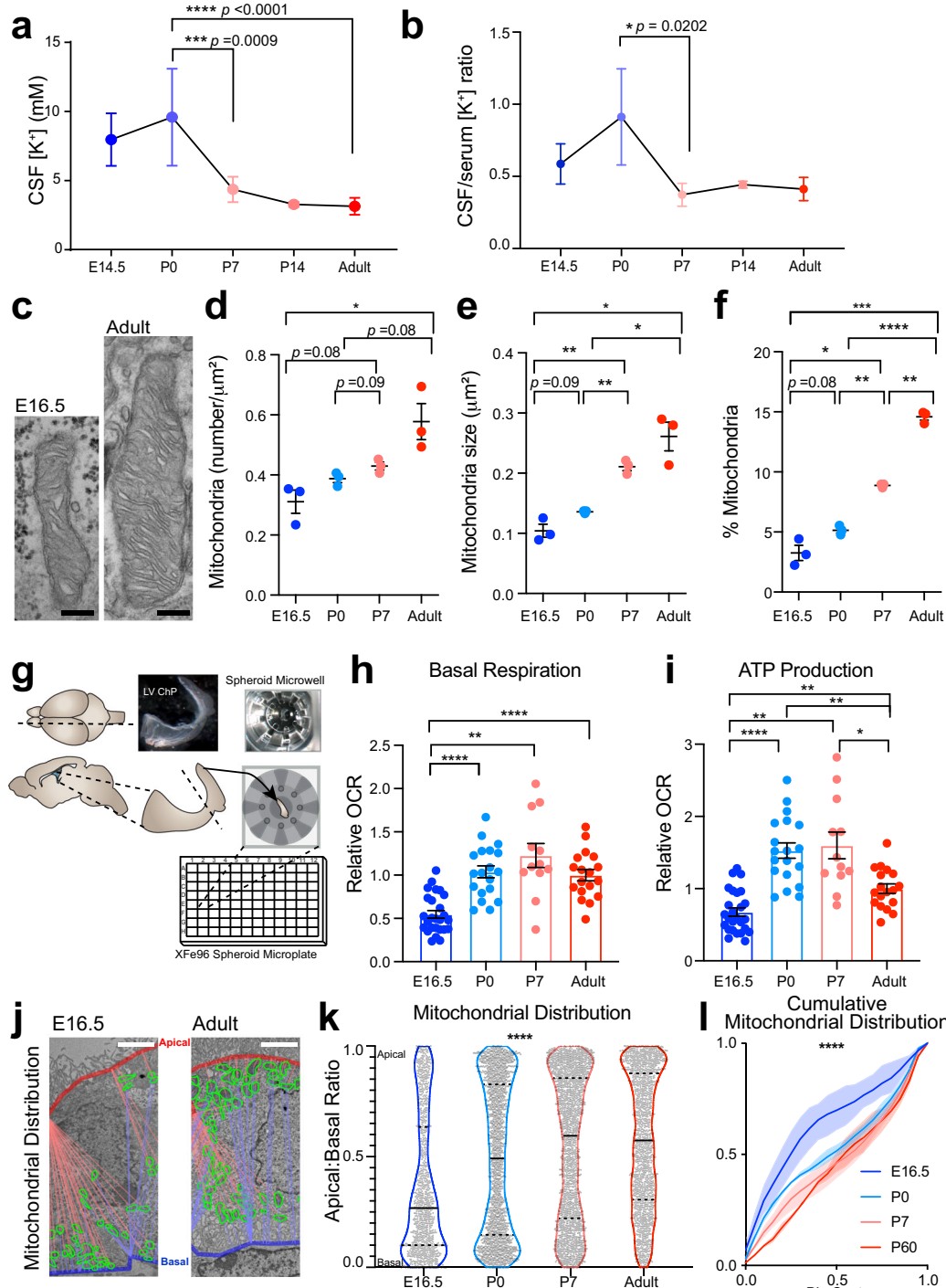

**Fig. 1 Postnatal CSF [K$^+$] decrease coincides with increased choroid plexus metabolism. a** ICP-OES quantification of CSF [K$^+$]. Colors: blue-to-red indicate increasing age in all panels. ***$p = 0.0009$, ****$p < 0.0001$; Sidak's test. **b** Developmental CSF/Serum [K$^+$] ratios. Ratio = CSF [K$^+$]/average serum [K$^+$]; Sidak's test. *$p = 0.0202$. **c** Transmission micrographs of LVChP mitochondria. **d–f** Quantification of mitochondrial number (**d**), area (**e**), and % area occupancy (**f**) in ChP epithelial cells. **d** *$p = 0.0269$; **e** E16.5 vs. P7 **$p = 0.0025$; E16.5 vs. adult *$p = 0.0115$; P0 vs. P7 **$p = 0.0062$; P0 vs. adult *$p = 0.0344$; **f** E16.5 vs. P7 *$p = 0.0112$; E16.5 vs. adult ***$p = 0.0008$; P0 vs. P7 **$p = 0.0010$; P0 vs. adult ****$p < 0.001$; P7 vs. adult **$p = 0.0010$; Welch's two-tailed unpaired $t$-test. **g** Schematic of explant-based Agilent Seahorse XFe96 test. **h, i** Oxidative respiration metrics over development; OCR oxygen consumption rate. **h** E16.5 vs. P0 ****$p < 0.0001$; E16.5 vs. P7 ***$p = 0.0025$; E16.5 vs. Adult ****$p < 0.0001$; **i** E16.5 vs. P0 ****$p < 0.0001$; E16.5 vs. P7 **$p = 0.0022$; E16.5 vs. Adult **$p = 0.0037$; P0 vs. Adult **$p = 0.0012$; P7 vs. Adult *$p = 0.0480$; Welch's ANOVA with Dunnett's T3 multiple comparison test. **j, k** Mitochondrial distribution apical: basal proximity ratio: 1 = apical surface. 0 = basal surface. Solid line: median; dashed line: upper/lower quartiles. ****$p < 0.0001$; Kolmogorov–Smirnov test. **l** Cumulative distribution of mitochondrial localization. Solid lines: mean; shaded area: range. Scale bar = **c** 250 nm, **j** 2 μm. All quantitative data presented as mean ± SEM. LVChP lateral ventricle choroid plexus, E embryonic day, P postnatal day. Information on replicates and reproducibility for this figure can be found in the "Statistics and Reproducibility" section of the Methods. Source data are provided as a Source Data file.

production (Fig. 1h, i). In addition, mitochondrial subcellular distribution in ChP epithelium was biased toward the apical surface as postnatal development proceeded, with E16.5 mitochondria heavily distributed along the basal side of epithelial cells, P0 mitochondria intermediately localized, and P7 and adult mitochondria having more apical distribution (Fig. 1j–l, Supplementary Fig. 3). Mitochondrial subcellular localization responds to regional energy demand in other cellular processes, such as migration of mouse embryonic fibroblasts and during axonal outgrowth[36,37]. Together with the increase in ATP production postnatally, the shift in ChP epithelial cell mitochondrial distribution over postnatal development suggests increasing ATP supply to meet high demand at the apical ChP surface during the early postnatal phase, concurrent with the rapid clearance of CSF $K^+$. Although many processes at the ChP apical surface consume ATP and could drive the mitochondrial distribution shift during this phase, this concurrence prompted us to investigate mechanisms whereby ChP epithelial cells might contribute to $K^+$ clearance through ATP-dependent mechanisms.

**The ChP increases production of CSF-facing ion and water transporters postnatally.** Consistent with rapid CSF $K^+$ clearance and high ChP metabolism, we found that expression of energy-dependent cation transport pathway components was upregulated in ChP postnatally. To identify which of the ChP transporters are likely candidates controlling postnatal CSF clearance through the ChP, we conducted ribosomal profiling to investigate transcripts that are prioritized for translation in embryonic (E16.5) and adult ChP, using Translating Ribosomal Affinity Purification (TRAP[38]). ChP epithelial cells were targeted by crossing *FoxJ1:cre* mice[39] with *TRAP* (*EGFP:L10a*) mice[38] (Fig. 2a, Supplementary Fig. 4a, b), and mRNA associated with the L10a ribosomal subunit were purified for sequencing.

TRAP analyses revealed 1967 differentially translated transcripts (adjusted $p < 0.05$) between E16.5 and adult ChP: 1119 enriched at E16.5 and 847 enriched in adults (Fig. 2b). Gene set and pathway analyses revealed developmentally regulated ChP programs. Adult ChP had enriched functional gene sets associated with active transmembrane membrane transport and mitochondria, which is consistent with our abovementioned findings on metabolism changes (Fig. 2c). Notably, cation transport was enriched, supporting the hypothesis of ChP mediating CSF $K^+$ transport postnatally (Fig. 2c, d, and Supplementary Data 1). Enriched pathways in the adult included secretion associated pathways named for other, better-studied secretory processes, including salivary and pancreatic secretion, all of which have a special emphasis on water and ion transmembrane transport (Supplementary Fig. 4c (red), d). Consistent with a rise in fluid and ion modulating machinery, there was a striking enrichment of more transmembrane and signal peptide-containing transcripts in adult ChP (Supplementary Fig. 4e, f, and Supplementary Data 2). In contrast, the water channel AQP1, which is not directly responsive to any given ion gradient, but rather depends on the overall osmolarity, remained unchanged (Supplementary Fig. 4h). These results indicate that the ChP gained fluid and ion modulatory functions postnatally, with an emphasis on ion-related transporters and channels.

**NKCC1 is poised to mediate perinatal ChP CSF $K^+$ and water clearance.** Among all fluid and ion modulating candidates with increasing postnatal expression (Supplementary Fig. 4g, h), we identified NKCC1 (*Slc12a2*) as the candidate most likely to mediate CSF clearance. NKCC1 is functionally related to $Na^+$/$K^+$-ATPase (*Atp1a1* and *Atp1b1*), as the latter actively maintains the $Na^+/K^+$ gradient that partially drives NKCC1 directionality.

Both the $Na^+/K^+$-ATPase and NKCC1 are capable of CSF $K^+$ clearance, but NKCC1 was of particular interest because (1) it is a cotransporter of $K^+$ and water[30,40]; and (2) the activity of NKCC1 can be further modified by phosphorylation[41], lending additional control over its fluid/ion modulatory capacity. In addition, NKCC1 is particularly enriched in the ChP and does not impact broad functionality like the $Na^+/K^+$-ATPase does, both of which are important features for a functional therapeutic intervention target. We refined our temporal expression analyses of NKCC1, ATP1a1, ATP1b1, and Klotho (Kl), which contributes to the membrane localization of the $Na^+/K^+$-ATPase[42,43] (Fig. 2e), by sampling weekly from P0 to P28 and then from adult, and confirmed increased expression of transcript and protein for each component across developmental time (Fig. 2f, g, Supplementary Fig. 5). The observed changes in NKCC1 total protein were corroborated by an independent approach where the rate of ChP epithelial cell swelling under high $[K^+]$ challenge[40] reflected the abundance of NKCC1 protein (Fig. 2h–j, Supplementary Fig. 6). Artificial CSF (aCSF) recipes that reflect the $[K^+]$ of neonatal vs. adult CSF were used for respective ages (see Supplementary Table 1 for recipes). Immunolabeling of NKCC1 showed its apical membrane localization at all ages, regardless of abundance (Supplementary Fig. 7).

In addition, we found particularly high levels of phosphorylated, therefore activated[41], NKCC1 (pNKCC1) in the ChP of P0-P7 pups, with P7 having peak pNKCC1 levels among all postnatal ages, indicative of increased NKCC1 activity during the first postnatal week (Fig. 2g). Similar to the timeline of ChP ATP production (Fig. 1i), the timeframe of high ChP pNKCC1 was concurrent with the fast CSF $[K^+]$ decrease during the first postnatal week (Fig. 1a, b), suggesting a functional correlation and further confirming the significance of the early postnatal transitional period. Taken together, we identified ChP NKCC1 as the top candidate for mediating postnatal CSF $K^+$ and water clearance.

**NKCC1 temporal regulation requires epigenetic control that is implicated in congenital hydrocephalus.** We found that the temporal profile of NKCC1 expression was tightly regulated at the epigenetic level by modulators implicated in some forms of congenital hydrocephalus. The NuRD complex governs differentiation and maturation of diverse cells and tissues[44]. Our previously published RNA sequencing studies[45] identified NuRD components, including the ATPase CHD family members (Chd4 being the most highly expressed), the histone deacetylases HDAC1/2, and methyl CpG-binding domain protein MBD3 in the ChP (Fig. 3a). De novo loss-of-function CHD4 mutations are implicated in some groups of children with congenital hydrocephalus and ventriculomegaly[46]. We found that CHD4 localized to nuclei in mouse ChP epithelial cells beginning at P0 (Fig. 3b). Immunoprecipitation of CHD4 identified HDAC1, HDAC2, and MBD3 by immunoblotting in mouse ChP (Fig. 3c, technical control for Co-IP protocol is shown in Supplementary Fig. 8), confirming the existence of the CHD4/NuRD complex in developing ChP. We then disrupted the complex by generating ChP-*Chd4* deficient mice. Cre was expressed in *Chd4* floxed mice (*Chd4* fl/fl)[47] using an adeno-associated viral vector (AAV) with tropism for the ChP (AAV2/5)[48], delivered by in utero intracerebroventricular (ICV) injection at E14.5. *Chd4* transcript levels decreased to <50% by P7 (Fig. 3d). While CHD4 protein levels only substantially decreased by P14 (Fig. 3e, g), we found that the developmental increase of ChP NKCC1 expression was disrupted as soon as the CHD4 protein decreased and lasted at least until P28 (Fig. 3f, g). Similar results were also observed in 4VChP (Fig. 3h, i). These data confirm that the NuRD/ChD4 complex is

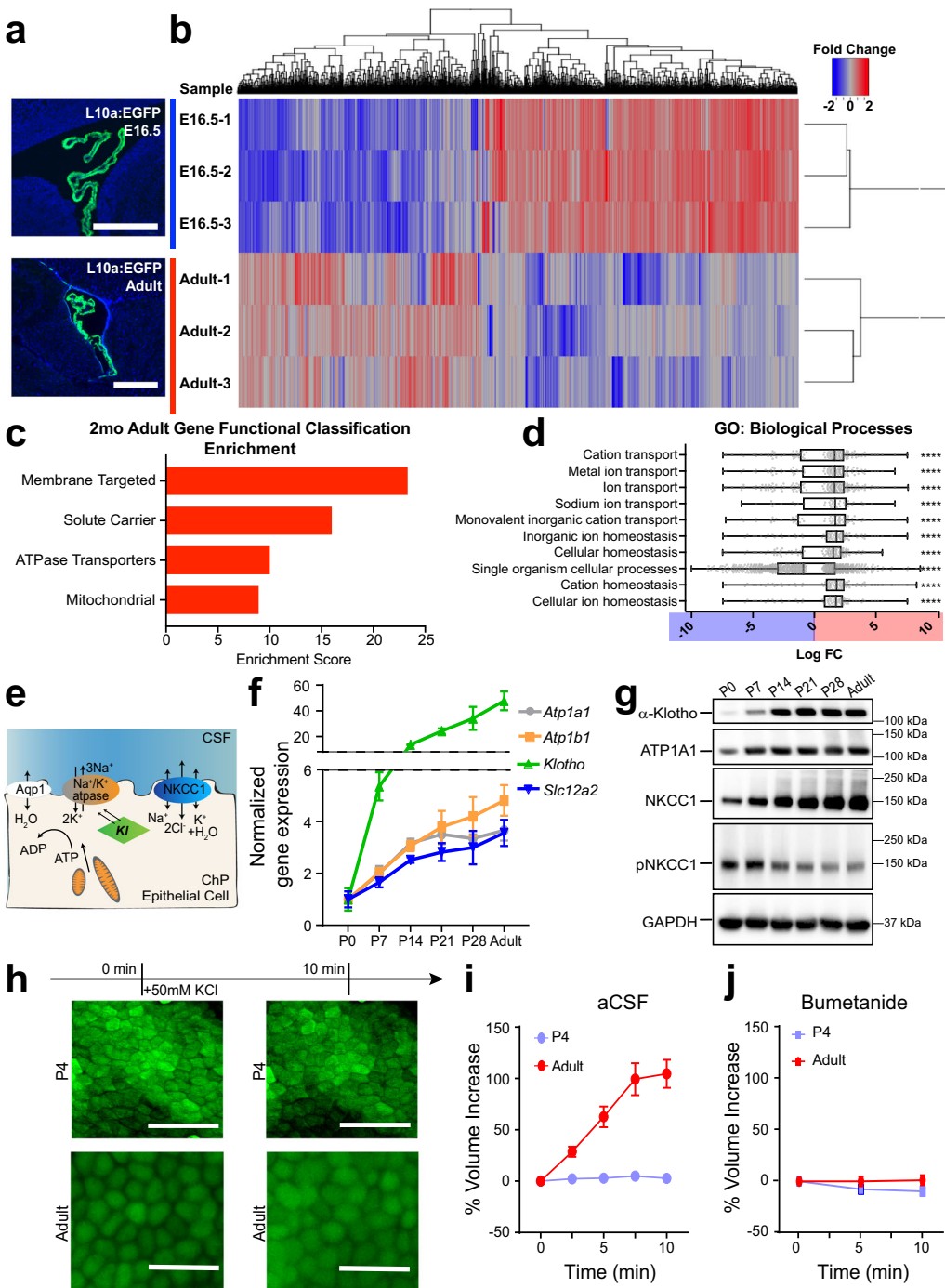

one of the required components tightly regulating ChP NKCC1 developmental expression.

**ChP NKCC1 actively mediates CSF clearance during the early postnatal transition phase.** To test whether NKCC1 is indeed capable of transporting from CSF into the ChP during the period of rapid CSF [$K^+$] decline, we induced NKCC1 overexpression (OE) in developing ChP epithelial cells using AAV2/5. NKCC1 transport directionality follows the combined $Na^+$, $K^+$, $Cl^-$ gradients, which are close to neutral in adult brains and likely to bias toward the CSF-to-ChP direction during the early postnatal phase. NKCC1 protein level would be rate-limiting during the early postnatal time when it is already highly phosphorylated,

unlike in older mice where pNKCC1 only represented a small portion of total NKCC1. The goal of this OE approach was to accelerate endogenous ChP NKCC1 transport, thereby revealing its directionality based on whether CSF [$K^+$] clearance was enhanced or delayed. AAV2/5-NKCC1, which expresses NKCC1 fused to an HA tag[49], or control GFP virus was delivered by in utero ICV at E14.5. Successful NKCC1 OE and increased pNKCC1 was confirmed in ChP at P0 (Fig. 4a–d). Appropriate localization to apical membranes of epithelial cells, transduction efficiency, and tissue specificity were also validated (Fig. 4e–i, Supplementary Fig. 9a, b). Transcript levels of other $K^+$ transporters or channels did not change following AAV2/5-NKCC1 transduction (Supplementary Fig. 9c). Because CSF [$K^+$] sharply decreased from P0 to P7 (Fig. 1a, b), we sampled CSF from ChP

**Fig. 2 Choroid plexus epithelial cells display age-dependent translation of ion and water transporters, in particular NKCC1. a** Rpl10a-conjugated EGFP expression in ChP epithelial cells after *Foxj1*-Cre recombination in TRAP-BAC mice. Scale bars = 500 μm. Representative of two experiments, each with two biologically independent replicates. **b** Heatmap and hierarchical clustering of differentially expressed genes (adjusted *p* < 0.05). Red: enriched adult expression. Blue: enriched E16.5 expression. **c** Top four gene functional clusters shown by DAVID to be enriched in Adult ChP epithelial cells over E16.5 ChP epithelial cells. **d** Top 10 significantly enriched gene ontology (GO) terms for "Biological processes". Plotted with horizontal lines for medians, bounds of boxes for quartiles, and whiskers for maximum and minimum values. The $\log_{10}$ fold change (LogFC) is plotted for each expressed gene for the network. Positive values (red): Adult enrichment; negative values (blue): E16.5 enrichment. Multiple measures were corrected using Bonferroni correction. **$p \leq$ 0.01, ***$p \leq 0.001$, ****$p \leq 0.0001$ (See Supplementary Data S1 for exact *p*-values). **e** Schematics depicting the interaction of NKCC1, $Na^+/K^+$-ATPase, and Klotho (Kl) on the apical membrane of a ChP epithelial cell. **f, g** RT-qPCR and immunoblotting of LVChP during postnatal development. **f** *N* = 4 biologically independent animals from two experiments at each timepoint. Colors are matched to the gene's protein name in Fig. 2e. g: representative of three independent experiments, each with tissues from 1–2 animals (two animals for ages under P14, one animal for ages of P14 and older) pooled for each timepoint. **h** Fluorescence images of Calcein-AM labeled epithelial cells from LVChP explants under high extracellular $K^+$ challenge· Scale bar = 50 μm, representative of four biological replicates collected from three independent experiments. Biological replicates with poor calcein labeling or visible damage were excluded prior to $K^+$ challenge. **i, j** Quantification of ChP epithelia cellular volume by IMARIS 3D analysis. Percent volume increase = $dV/V_0$ for each timepoint (t). $V_0$ = initial volume of the cell; t = subsequent timepoint after challenge; $dV = V_t - V_0 \times 100\%$. At least five cells were analyzed for each explant from each animal; *N* = 4 animals. Red: adults; light blue: P4. All quantitative data are presented as mean ± SD. Source data are provided as a Source Data file.

NKCC1 OE and control mice at P1. We found that ChP NKCC1 OE reduced CSF $[K^+]$ more than controls, with their P1 CSF $[K^+]$ values closely approximating those normally observed at P7 (Fig. 4j), indicating accelerated $K^+$ clearance from CSF after enhanced ChP NKCC1 activity. CSF total protein levels were not affected (AAV2/5-GFP = 2.50 ± 0.20 mg/ml vs. AAV2/5-NKCC1 = 2.71 ± 0.46 mg/ml; *N* = 6 from two litters each; *p* = 0.34, unpaired *t*-test). Overall, these findings support a model in which, under physiological conditions with high early postnatal CSF $[K^+]$, ChP NKCC1 transports $K^+$ out of the CSF.

Next, we found that the circulating CSF volume in ChP NKCC1 OE mice was reduced, as reflected by smaller lateral ventricles. To avoid any tissue processing artifacts, we conducted live T2-weighted magnetic resonance imaging (MRI) (Fig. 5a) to quantify lateral ventricle volume. AAV-GFP mice were indistinguishable from naive wild-type mice at P14. In contrast, NKCC1 OE mice had reduced lateral ventricle volumes (Fig. 5a, b), without decrease in overall brain size (Fig. 5c), reflecting less circulating CSF. The difference in ventricle sizes from these same mice was sustained when measured again at P50 (AAV-GFP: 3.12 ± 0.59 mm³ vs. AAV-NKCC1: 1.28 ± 0.28 mm³, *$p$* = 0.0182), suggesting ventricular deflation established early could dictate long-term changes in ventricle structure.

The observed lack of ventricle enlargement at a later stage is also consistent with the finding that only a small portion of NKCC1 was phosphorylated in adult ChP (Fig. 2g), and therefore overexpression without increased activation via phosphorylation was unlikely to influence total NKCC1 transport in adulthood. While the exact transport direction of NKCC1 in adult ChP is still under debate[24], the consistency in ventricular volume from P14 into later life supports our working model that because a relatively small proportion of ChP NKCC1 was phosphorylated in mice P14 and older (Fig. 2g), NKCC1 levels are not rate-limiting and thus OE would not as substantially impact ChP functions in older animals. Collectively, our findings demonstrate that ChP NKCC1-mediated CSF clearance during the first postnatal week. Augmenting this process impacted CSF volume homeostasis in the long term.

We then tested and found that enhancing CSF clearance through ChP NKCC1 OE changed how the brain and cranial space adapted to CSF volume changes. Intracranial compliance ($C_i$) and CSF resistance ($R_{CSF}$) describe the ability of the entire intracranial space (including brain, meninges, and outflow routes) to accommodate an increasing CSF volume that would otherwise increase intracranial pressure (ICP). In humans, these parameters are measured by a CSF constant rate infusion test[50–52], and can

aid in diagnosis and evaluation of conditions like hydrocephalus, which has decreased $C_i$[2]. ICP and $R_{CSF}$ have been previously measured in rats[53] using a similar test, quantifying ICP based on mmH₂O in a glass capillary. It was noted that $R_{CSF}$ changes with infusion rate in a non-linear way, and $R_{CSF}$ was estimated based on linear parts of the curve each representing high or low infusion rates. We developed a miniaturized version of the human test with a pressure sensor coupled with infusion tubing implanted inside the lateral ventricles to determine the $C_i$ and $R_{CSF}$ in mice. The constant rate infusion test artificially increases CSF volume by ICV infusion of aCSF, causing ICP to rise and plateau at a new level (Fig. 5d, e). The $C_i$ and $R_{CSF}$ are estimated from the ICP vs. time curve using Marmarou's model of CSF dynamics[54] (Supplementary Fig. 10a). Simply put, the $C_i$ is inversely proportional to the rate of ICP increase, and the $R_{CSF}$ is related to the level of the post-infusion ICP plateau (Fig. 5e). As a quality control for the correct placement of infusion and measurement catheter, arterial and respiratory pulsations were clearly visible in the ICP waveform and their amplitude increased with volume load as expected (Supplementary Fig. 10b, c). Using this approach, we found that ChP NKCC1 OE significantly increased $C_i$ at an age of 5–7 weeks (Fig. 5f, g), consistent with the brain having greater capacity for CSF in ventricles "deflated" due to excessive CSF clearance. Resting ICP and $R_{CSF}$ were unchanged (Fig. 5h, i).

**Enhanced ChP NKCC1 function mitigates ventriculomegaly in a model of obstructive hydrocephalus.** Our findings of enhanced CSF clearance after ChP NKCC1 OE indicate that ChP NKCC1 can remove excess CSF. Therefore, we hypothesized that ChP NKCC1 OE expression could mitigate ventriculomegaly in a model of postnatal obstructive hydrocephalus. We first over-expressed ChP NKCC1 at E14.5 by in utero AAV2/5 ICV, then introduced obstructive hydrocephalus by a single unilateral injection of kaolin into the lateral ventricle at P4[55], and finally evaluated the lateral ventricle volumes by live T2 MRI at P14 (Fig. 6a). While both NKCC1 OE and control mice had enlarged ventricles at P14, NKCC1 OE mice had reduced ventriculomegaly compared to controls, with the average ventricle volume being less than 1/3 of the controls (Fig. 6b–d; ventricles marked by blue arrows; kaolin deposits marked by red arrows). Taken together, our findings demonstrate that early, ChP targeted NKCC1 OE has a sustained and broad impact on specific volumetric and biophysical parameters of the intracranial space with potential therapeutic applications to hydrocephalus.

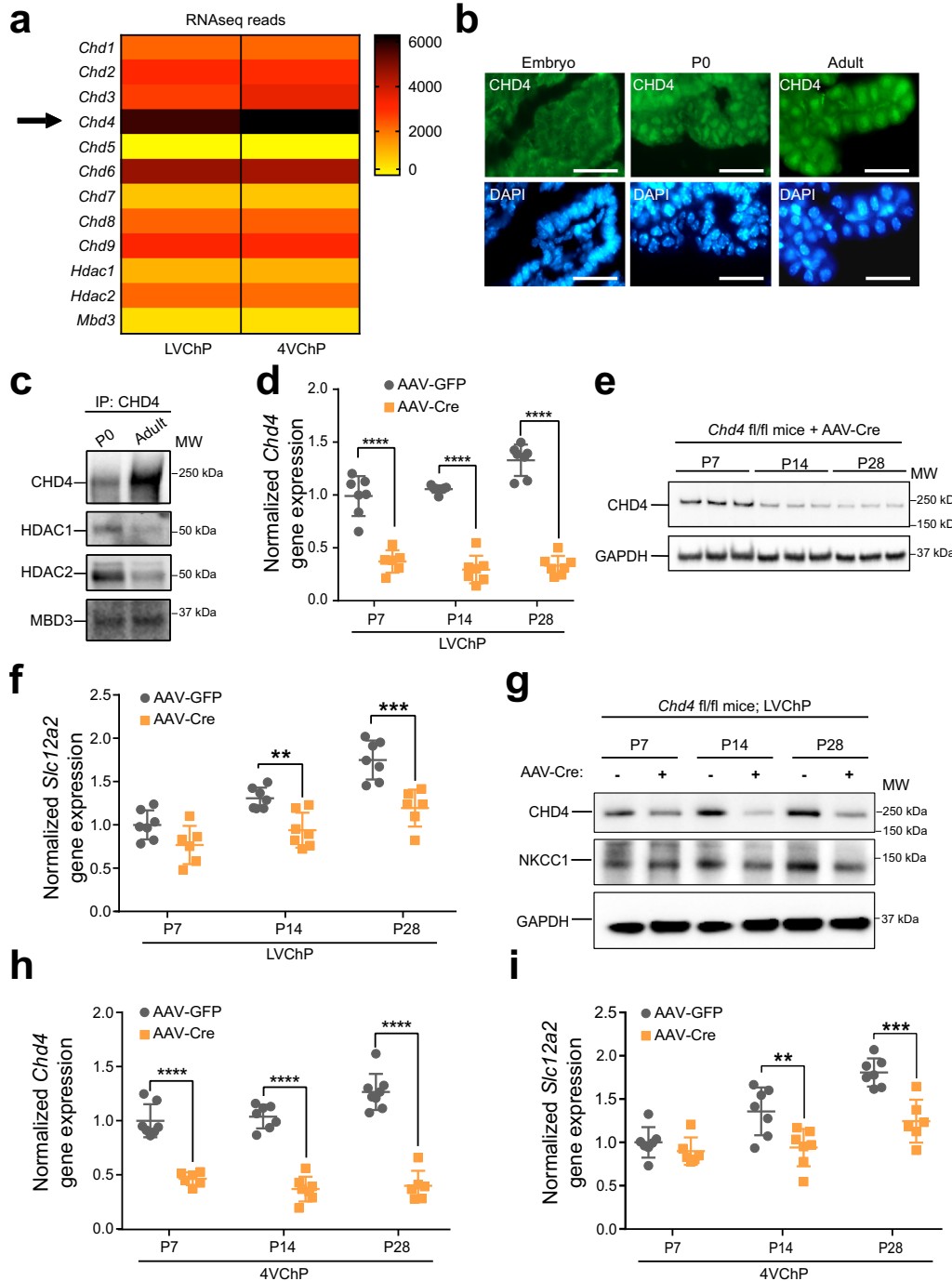

**Fig. 3 NKCC1 temporal expression requires the CHD4/NuRD complex. a** RNA-seq data showing expression of CHD and other NuRD units by the ChP. Yellow: low expression; Dark brown: high expression. **b** Immunofluorescence images of CHD4 in the ChP epithelia at E16.5, P0, and adult; Scale bar = 30 μm. A total of four animals at each age were imaged in two independent experiments. **c** Immunoblots of Co-IP by CHD4 antibody. Representative of two independent experiments, each contain more than 10 animals for each age. **d** RT-qPCR of CHD4 transcripts in ChP with AAV2/5-Cre transduction. Gray: AAV-GFP; orange: AAV-Cre (same color scheme is used for the rest of the figure). ****$p < 0.0001$, $N = 7$, Welch's two-tailed unpaired $t$-test. **e** Immunoblot of CHD4 in AAV-Cre mice ChP lysate, representative of three independent experiment, each contain two or three independent animals. **f** RT-qPCR of NKCC1 expression in AAV-Cre vs. AAV-GFP mice ChP. All values were normalized to P7 AAV-GFP control mice. **$p = 0.0015$, ***$p = 0.0009$, $N = 7$, Welch's two-tailed unpaired $t$-test. **g** Immunoblot of NKCC1 in LVChP lysates from AAV-Cre vs. AAV-GFP mice, representative of three independent experiments, each containing two or three independent animals (same samples as those collected for Fig. 2e). **h, i** CHD4 and NKCC1 RT-qPCR in 4VChP. **$p = 0.0083$, ***$p = 0.0005$, ****$p < 0.0001$, $N = 7$, Welch's two-tailed unpaired $t$-test. All quantitative data are presented as mean ± SD. When immunoblots were quantified, all samples for quantitative comparison were on the same blot. Source data are provided as a Source Data file.

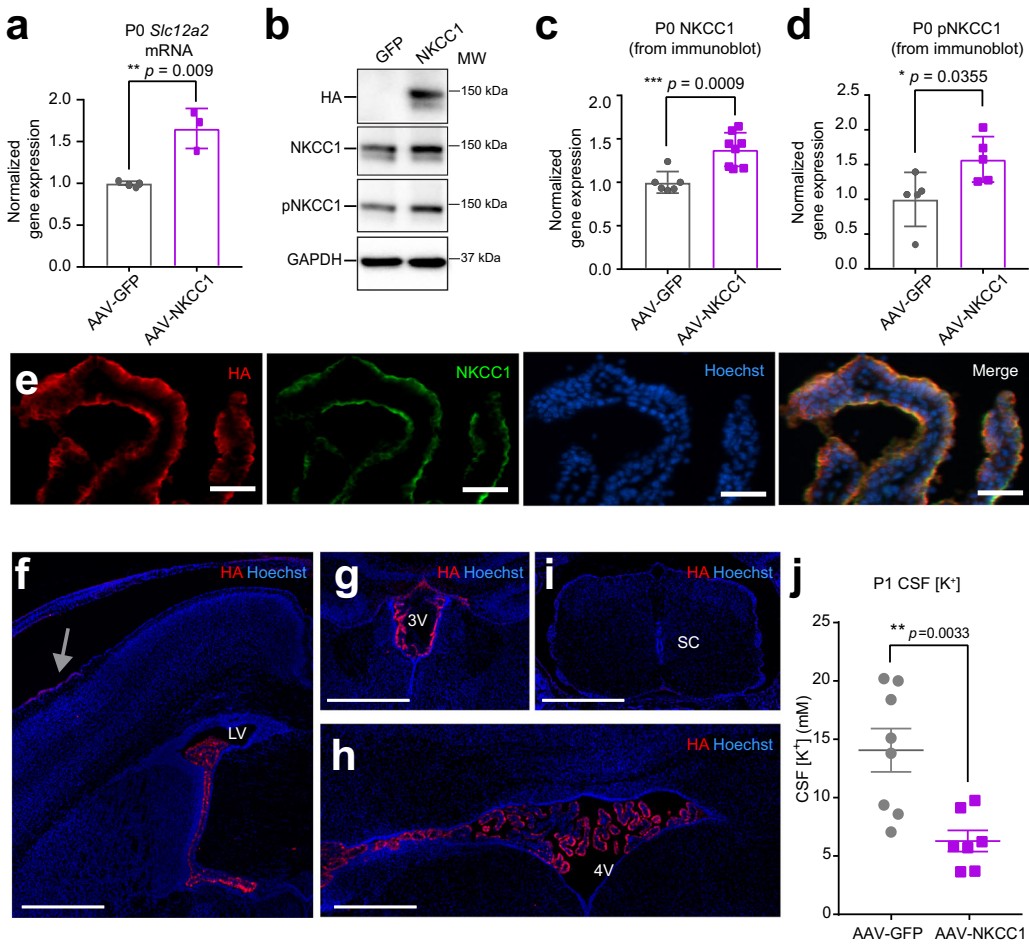

**Fig. 4 ChP NKCC1 actively mediates CSF K$^+$ clearance during the first postnatal week. a** RT-qPCR of NKCC1 mRNA levels in P0 mice. Gray: AAV-GFP; purple: AAV-NKCC1 (same color scheme is used for the rest of this figure). **$^{**}p = 0.009$, $N = 3$; Welch's two-tailed unpaired $t$-test. **b** Immunoblots from AAV-NKCC1 vs. AAV-GFP P0 mice ChP lysates, representative of three independent experiments, each contained multiple biological replicates (the exact numbers and raw blots are included in the Source Data with quantification presented in panels **c** and **d**). **c, d** Quantification of all immunoblots of NKCC1 **c** $^{***}p = 0.0009$, $N = 7$; Welch's two-tailed unpaired $t$-test; and pNKCC1 **d** $^{*}p = 0.0355$, $N = 5$; Welch's two-tailed unpaired $t$-test. All samples for direct quantitative comparison were on the same blot (see Source Data). **e** Immunofluorescence images showing colocalization of 3xHA tag and NKCC1 in P0 ChP. Scale bar = 50 μm, representative of three independent experiments, each with two biological replicates. **f–i** Immunofluorescence images of HA in AAV2/5-NKCC1 transduced brain at P1: LVChP (**f**), 3$^{rd}$ ventricle ChP (3VChP) (**g**), 4VChP (**h**), and the spinal cord (**i** sc = spinal cord). Trace expression of HA in the meninges near the injection site (gray arrow). Scale bar = 500 μm. **f–i** are representative of two independent experiments, each with three biological replicates. **j** ICP-OES measurements of CSF [K$^+$] from AAV-NKCC1 vs. AAV-GFP P1 mice ($N = 8$ in AAV-GFP cohort; $N = 7$ in AAV-NKCC1 cohort). $^{**}p = 0.0033$; Welch's two-tailed unpaired $t$-test. All quantitative data are presented as mean ± SD. Source data are provided as a Source Data file.

## Discussion

In this study, we sought to understand how CSF is cleared from the brain during a period of development prior to the maturation of CSF outflow routes suggested to exist in adults (e.g., arachnoid granulations, arachnoid villi, perineural and paravascular pathways, and meningeal lymphatics)[1,8,11–14,56]. Failure of CSF clearance during this developmental period has debilitating consequences[7]. Our results suggest that this period is defined by a rapid decrease in CSF K$^+$. The ChP mediates CSF K$^+$ clearance during this transition period, and thus forms a CSF outflow route through ion and water co-transport by NKCC1 (Fig. 7). This CSF clearance by the ChP during normal development contrasts with the prevailing view that healthy ChP always mediates net secretion (rather than absorption) of CSF. The prospect that the ChP may in different contexts absorb or secrete CSF has been previously hypothesized to occur under both physiological and disease conditions[15,19–22]. Thus, our findings demonstrate a precisely timed function of the developing ChP in CSF

clearance prior to the formation of other routes, and provide targets for fluid management intervention in hydrocephalus.

NKCC1 is a bidirectional transporter, recently shown to be an important cotransporter of water in the adult ChP[40]. Although established as a key molecular mediator of CSF regulation, ChP NKCC1 transport direction and its determinants in vivo have been debated due to the technical challenges of (1) specifically manipulating ChP NKCC1 without affecting NKCC1 in other CSF-contacting cells (ICV application of NKCC1 inhibitors such as bumetanide suffer from this limitation); and (2) accurately determining intracellular ion levels of ChP epithelial cells, and therefore ion gradients, under physiological conditions, as summarized in Supplementary Table 2 and reviewed by Delpire and Gagnon[24]. Our in vivo "gain-of-function" approach effectively bypasses the abovementioned technical limitations. By overexpressing NKCC1 specifically in the ChP, we could subsequently observe the resulting CSF K$^+$ and fluid volume changes that revealed the transporter's net directionality. Using this approach,

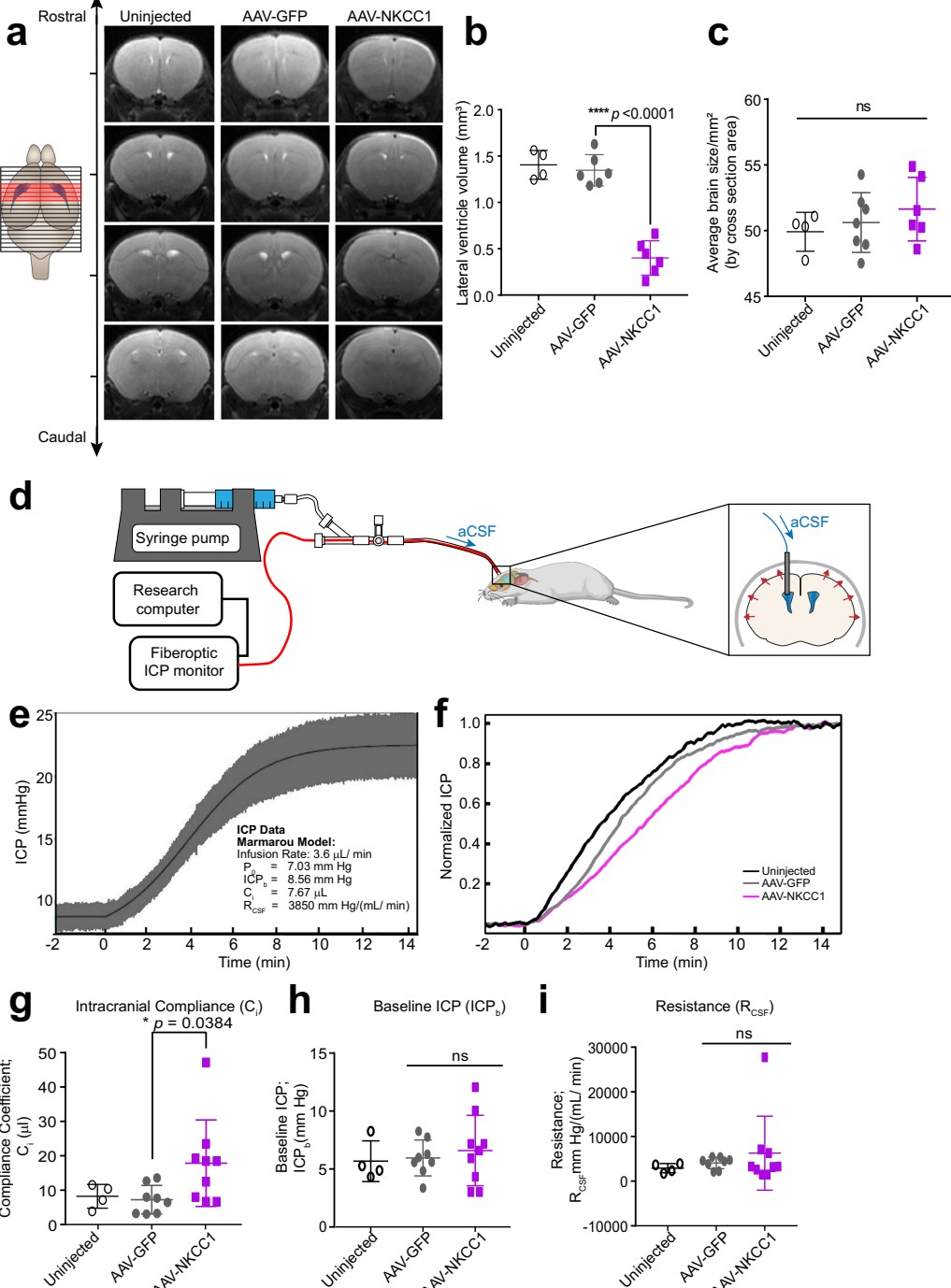

**Fig. 5 ChP NKCC1 overexpression reduces brain ventricular volume and increases intracranial compliance. a** T2-weighted live MRI images. Only slices with visible lateral and 3rd ventricles are shown (marked red in the schematics; for NKCC1 OE mice, slices matching with control mice are shown regardless of ventricles). **b** Lateral ventricle volumes. Uninjected $N = 4$; AAV-GFP and AAV-NKCC1 $N = 6$, from two independent experiments; Black with no fill: uninjected; gray: AAV-GFP; purple: AAV-NKCC1 (the same color scheme is used for the rest of this figure). ****$p < 0.0001$; Welch's two-tailed unpaired $t$-test. **c** Brain sizes, which are presented as the average coronal section area from all images with visible lateral and 3rd ventricles (marked red in the schematic; NKCC1 OE data were calculated using the matching images to the controls, regardless of ventricles visibility). Uninjected $N = 4$; AAV-GFP and AAV-NKCC1 $N = 6$ (same as mice included in panel **b**); Welch's two-tailed unpaired $t$-test. **d** Schematic of in vivo constant rate CSF infusion test. **e** Example of ICP curve during the infusion test (infusion begins at 0 min) in an AAV-GFP mouse, fitted to Marmarou's model. Values extracted include: baseline ICP (ICP$_b$), a pressure-independent compliance coefficient (C$_i$) and the resistance to CSF outflow (R$_{CSF}$). **f** Example ICP recordings from AAV-NKCC1 mice and controls. For clarity, data have been low pass filtered to remove the waveform components. **g** Compliance coefficients. Uninjected $N = 4$; AAV-GFP $N = 8$; AAV-NKCC1 $N = 9$; 3 total independent experiments. *$p = 0.0384$; Welch's two-tailed unpaired $t$-test. **h, i** Plots of baseline ICP and resistance to CSF outflow (R$_{CSF}$) at 5–7 weeks. Uninjected $N = 4$, AAV-GFP $N = 8$, AAV-NKCC1 $N = 9$; 3 total independent experiments (same experiments as those included in **g**). Welch's two-tailed unpaired $t$-test. All quantitative data are presented as mean ± SD.

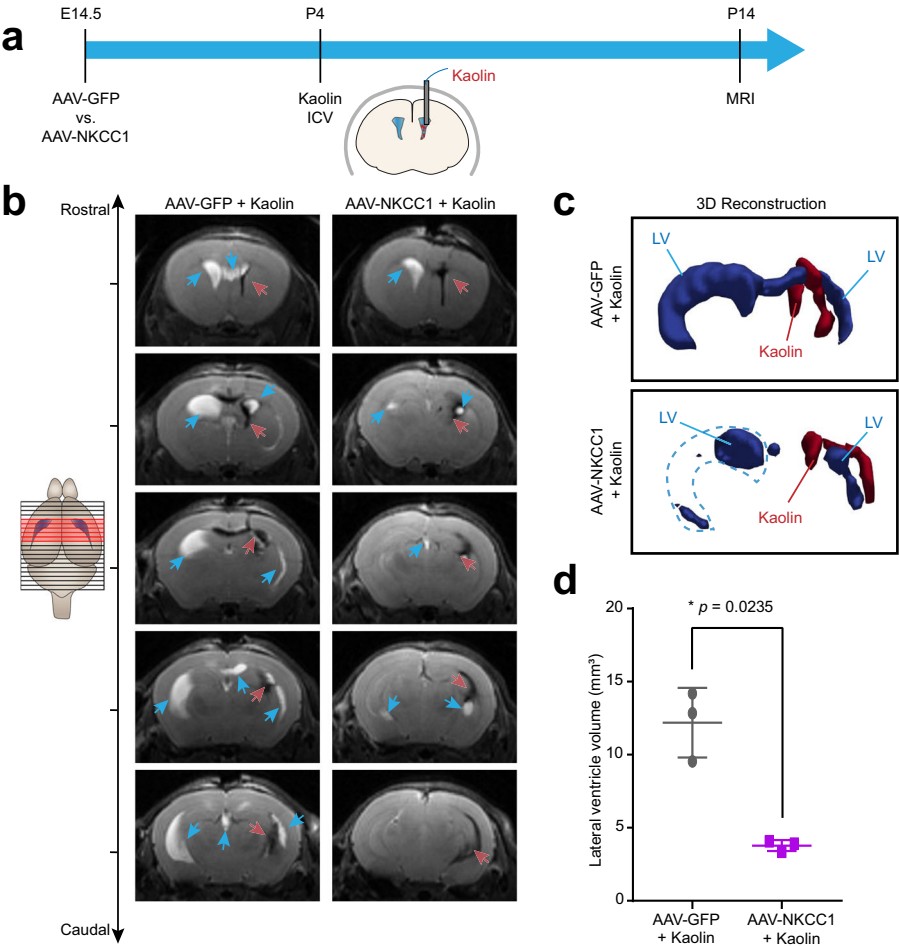

**Fig. 6 ChP NKCC1 overexpression mitigates ventriculomegaly in an obstructive hydrocephalus model. a** Schematics showing the workflow: E14.5 in utero ICV of AAV2/5-NKCC1 or AAV2/5-GFP, followed by ICV of kaolin at P4, and MRI at P14. **b** Representative sequential brain images (rostral to caudal) by T2-weighted live MRI images. Blue arrows: Lateral ventricle (LV). Red arrows: kaolin. **c** 3D reconstruction of the LV and kaolin deposition. LV: blue. Kaolin: red. **d** Lateral ventricle volumes. $N = 3$ from two biologically independent litters under each condition; Gray: AAV-GFP mice with kaolin; purple: AAV-NKCC1 mice with kaolin. *$p = 0.0.0235$; Welch's two-tailed unpaired $t$-test. All quantitative data are presented as mean ± SD.

we found that, in contrast to the common notion that the ChP constantly produces CSF under physiological conditions, ChP-NKCC1 mediates CSF clearance when normal CSF [K$^+$] is above adult values, especially during the first postnatal week in mice. Future studies should shed light on how this critical period corresponds to the human developmental timeframe, and whether shared mechanisms during development underlie vulnerability to disorders involving CSF.

We next demonstrated that ChP clearance of CSF can be targeted to temper abnormal CSF accumulation. CSF adjacent tissues, including ChP, have been targeted for therapeutic manipulation in rodent models of neurologic diseases ranging from Huntington's disease and lysosomal storage disorders, to Alzheimer's disease, where transduction of exogenous gene products has improved cardinal features of disease[48,57–60]. Here, enhancing ChP epithelial cell NKCC1 transport capacity lessened the severity of ventriculomegaly in a model of obstructive hydrocephalus. Our data suggest the possibility of treating congenital hydrocephalus by augmenting endogenous ChP NKCC1 activity to increase CSF absorption rates during early development when CSF [K$^+$] is high. In adult animals, CSF K$^+$ is rapidly removed when the concentration exceeds homeostatic levels[28,29]. It is tempting to speculate that depending on circumstances, ChP NKCC1 may provide a route for CSF K$^+$ clearance in adults as well.

In light of recent findings reporting hydrocephalus and ventriculomegaly in children with de novo loss-of-function CHD4 mutations[46], we found that the CHD4/NuRD complex is required for developmental regulation of NKCC1 expression. This connection suggests a possible pathophysiological mechanism whereby lack of CHD4 activity may reduce NKCC1 levels during early development (equivalent to P0-P7 in mice), and lead to insufficient CSF clearance resulting in hydrocephalus. In our loxP-cre approach, most CHD4 protein knockdown and resulting attenuation of NKCC1 expression occurred by P14, which is beyond the critical window of high NKCC1 activity observed between P0-P7. Improved genetic tools for even earlier CHD4 knockout and new animal models harboring the de novo patient mutations would be required to fully unravel the regulatory connection between the CHD4/NuRD complex and NKCC1.

A key question that emerges from this work is: If ChP NKCC1 mediates net CSF clearance during this transitional developmental stage, what other sources contribute to the influx of early CSF water and ions? In addition to possible ChP water secretion that is independent of NKCC1, forebrain progenitor cells that have a cell body at the ventricular surface and extend their basal processes to the developing pia could play a role in early CSF generation[61]. Consistent with progenitor involvement in CSF dynamics, recently identified genes driving pediatric

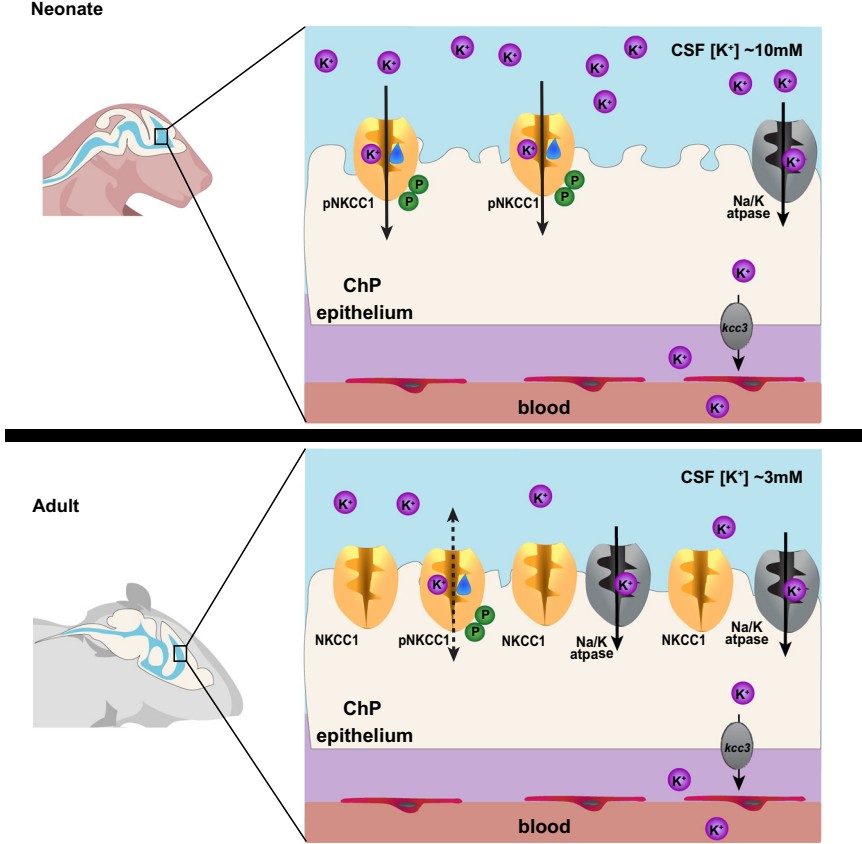

**Fig. 7 Working model of ChP NKCC1 mediating K⁺-driven CSF outflow.** The schematics depict ChP NKCC1-mediated K$^+$ and water clearance from CSF in neonatal mice, in comparison to the adult scenario. For simplicity and clarity, only K$^+$ is depicted among all ions and only NKCC1 and Na$^+$/K$^+$-ATPase are included. Neonatal (P0-7, above) ChP has higher pNKCC1 expression than adult ChP, albeit lower total NKCC1. Neonatal CSF [K$^+$] is 2–3 fold higher than adult. With similar [Na$^+$] and [Cl$^-$], this [K$^+$] difference is sufficient to alter the total Nernst potential of epithelial cells and bias NKCC1 transport of K$^+$, together with water, out of CSF and into the ChP in neonates.

hydrocephalus are expressed predominantly by cortical progenitor cells[62], and not by the ChP.

In addition to fluid regulation, our newly identified ChP clearance route provides a means for regulating extracellular K$^+$ during the first postnatal week. Ionic homeostasis of the CSF plays critical roles in brain development and function. The enzyme that moves K$^+$ against its individual concentration gradient, the Na$^+$/K$^+$-ATPase (i.e., Atp1a1 and Atp1b1 subunits), is not yet expressed at adult levels during this period. The ChP NKCC1-mediated K$^+$ clearance mechanism might assist in establishing the K$^+$ gradient in a timely manner, which is crucial for cellular physiology[63]. Notably, the period of rapidly decreasing CSF [K$^+$] overlaps with the developmental phase when the excitatory-to-inhibitory "GABA switch" occurs. In early cortical progenitor cells that reside in the ventricular zone and are bathed by CSF, the classic inhibitory neurotransmitter GABA leads to excitatory potentials and suppression of DNA synthesis[64]. As newborn cortical neurons differentiate and migrate away from the ventricular zone, GABA adopts its classic role as an inhibitory neurotransmitter in the context of lower intracellular Cl$^-$ [65], which is achieved through coordinated activities of neuronal K$^+$/Cl$^-$ co-transporters KCC2 and NKCC1[66,67]. Because ions can traffic from CSF into interstitial fluid[68,69], any interference with the developmental timeline of ChP NKCC1 function that results in delayed CSF K$^+$ clearance could potentially increase extracellular/interstitial fluid [K$^+$] and affect global neural excitability[63]. Specifically, such a change in extracellular/interstitial fluid [K$^+$] could fundamentally impact neuronal NKCC1 and KCC2 transport equilibrium, potentially contributing to a delayed

GABA switch, a phenomenon reported in many models of neurodevelopmental and psychiatric disorders including subtypes of autism spectrum disorder[70], Rett syndrome[71], Fragile X syndrome[72], schizophrenia[73], and Down syndrome[74]. Furthermore, extracellular [K$^+$] and certain K$^+$ channels also regulate microglia[75], which are critical for synaptic pruning during postnatal development in mice[76]. Thus, the ChP is poised to coordinate proper CNS formation by maintaining an optimal extracellular environment at different developmental stages, with possible effects on neuronal maturation, circuit formation, and neuroimmune interactions. Similarly, since many experimental paradigms in neurobiology commonly utilize aCSF to mimic the extracellular fluid environment, it is important to note that CSF ionic composition changes substantially over the course of development. In the present study, we provide ionically adjusted aCSF recipes with Na$^+$, K$^+$, Ca$^{2+}$, Mg$^{2+}$, and Cl$^-$ values for multiple developmental stages including embryonic, neonatal, postnatal, and adult.

Beyond key findings and implications during this critical developmental stage, our study introduced a murine ICP measurement device combined with constant CSF infusion. This approach can be broadly applied to study the CSF system across the mouse lifespan. We adapted our tool from clinical practice to provide a range of options for measuring global cerebral fluid states that reflect the interaction between CSF and cranial tissues. In later life, CSF homeostasis is maintained by collaborative efforts from multiple putative players in the brain, including the ChP, the dural lymphatics[14], leptomeningeal vasculature[77], and the ependyma[78]. While this approach measures overall cranial

fluid dynamics as one single unit, future applications could apply mathematical models that have been proposed to isolate the contribution of distinct CSF outflow routes using data acquired from patients[79].

Our findings reveal that the ChP regulates CSF fluid volume and ion composition at an early stage, which may play a role in guiding early brain development. Targeting this absorption route holds promise for improving fluid management in congenital hydrocephalus and other brain disorders.

## Methods

**Mice**. The Boston Children's Hospital IACUC approved all experiments involving mice in this study. Timed pregnant CD1 dams were obtained from Charles River Laboratories. Mice with germline loxP-CHD4-loxP were imported from MGH and bred in-house. Both male and female mice were equally included in the study and were analyzed at postnatal day 0, 7, 14, 21, 28, 5–7 weeks, and 2+ months. Animals were housed in a temperature and humidity controlled room on a 12-h light/12-h dark cycle and had free access to food and water. All mice younger than postnatal day 10 were allocated into gestational/postnatal age groups without respect to sex (both males and females were included). We intentionally included both males and females in studies involving mice older than 10 days.

**CSF collection and metal detection**. CSF was collected by inserting a glass capillary into cisterna magna, and collected CSF was centrifuged at $10,000 \times g$ for 10 min at 4 °C to remove any tissue debris. Metal quantification was performed by Galbraith Laboratories, Inc (Knoxville, TN, USA). Inductively coupled plasma–optical emission spectrometry (ICP-OES) was used for $K^+$ and $Na^+$ quantification, and ion chromatography (IC) was used for the $Cl^-$ quantification. All tests were performed using 5–10 μL of CSF. The quality control (QC) recovered at 94.4% for $K^+$, 97.7% for $Na^+$, 100.1% for $Cl^-$, 95.1–118% for $Mg^{2+}$, and 107.5 % for $Ca^{2+}$.

**TRAP**. We used a total of 9 mice aged 8 weeks and 9 litters of mouse embryos aged E16.5 from the Foxj1:Cre × EGFP-L10a Bacterial Artificial Chromosome (BAC) transgenic lines ($N = 3$ samples of LVChP pooled from 3- to 8-week-old mice or 3 litters of E16.5 embryos). Brain tissue was immediately dissected and used for TRAP RNA purifications[38]. ChP tissues were dissected in ice-cold dissection buffer (1 × HBSS; 2.5 mM HEPES-KOH; 35 mM glucose; 4 mM NaHCO3; 100 μg/mL CHX) and homogenized in lysis buffer (20 mM HEPES-KOH; 5 mM MgCl2; 150 mM KCl; 0.5 mM DTT; 1× protease inhibitors; 40 U/mL Rnasin; 20 U/mL Superasin) at 900 rpm for 12 strokes on ice with Teflon pestles in glass tubes (Kontes). Post-nuclear supernatant was prepared by centrifugation at 4 °C for 10 min at $2000 \times g$. Post-mitochondrial supernatant was prepared after incubation with additional 1% NP-40 and 30 mM DHPC, followed by centrifugation at 4 °C for 10 min at $20,000 \times g$. Immunoprecipitation was performed with magnetic streptavidin beads (MyOne T1 Dynabeads; Thermo Fisher #65601) conjugated to biotinylated anti-GFP antibodies (clones 19C8 and 19F7) for 16 h at 4 °C with gentle end-over-end rotation. Beads were collected on a magnet on ice, washed four times with 1000 μL 0.35 M KCl wash buffer (20 mM HEPES-KOH; 5 mM MgCl2; 350 mM KCl; 1% NP-40; 0.5 mM DTT; 100 μg/mL CHX). RNA was eluted in Stratagene Absolutely RNA Nanoprep Kit (Agilent #400753) lysis buffer (with β-ME) and purified according to kit instructions. RNA quality was assessed using Bioanalyzer Pico Chips (Agilent, 5067-1513) and quantified using Quant-iT RiboGreen RNA assay kit (Thermo Fisher Scientific R11490). Libraries were prepared using Clonetech SMARTer Pico with ribodepletion and Illumina HiSeq to 50NT single end reads. Sequencing was performed at the MIT BioMicroCenter.

**Sequencing data analysis**. The raw fastq data of 50-bp single-end sequencing reads were aligned to the mouse mm10 reference genome using STAR RNA-Seq aligner (v2.4.0j)[80]. The mapped reads were processed by htseq-count of HTSeq software (v 0.6.0)[81] with mm10 gene annotation to count the number of reads mapped to each gene. The Cuffquant module of the Cufflinks software (v 2.2.1)[82] was used to calculate gene FPKM (Fragments Per Kilobase of transcript per Million mapped reads) values. Gene differential expression test between different animal groups was performed using DESeq2 package (v. 1.26.0)[83] with the assumption of negative binomial distribution for RNA-Seq data. Genes with adjusted p-value < 0.05 are chosen as differentially expressed genes. All analyses were performed using genes with FPKM > 1, which we considered as the threshold of expression.

**Sequencing pathway and motif analysis**. Functional annotation clustering was performed using DAVID v6.7[84]. Gene ontology (GO) analysis was performed using AdvaitaBio iPathway guide v.v1702. Enrichment vs. perturbation analysis was performed by AdvaitaBio iPathway guide v.v1702 and allows comparison of pathway output perturbation and cumulative geneset expression changes. In brief, the enrichment analysis is a straightforward geneset enrichment over-representation analysis (ORA) considering the number of differentially expressed genes (DEGs) that are assigned to a given pathway. The enrichment value is expressed as a proportion of enriched members to total genes in a defined pathway and a p-value (Fisher) is calculated for this score; however, false positives have been reported at up to 10% with this method[85]. Perturbation, on the other hand, uses pathway data that applies relationships between gene products rather than only a list. Perturbation assigns an impact score based on a mathematical model that captures the entire topology of the pathway and uses it to calculate how changes in the expression of each gene in the pathway would perturb the absolute output of the pathway[85]. Then, these gene perturbations are combined into a total perturbation for the entire pathway and a p-value is calculated by comparing the observed value with what is expected by chance. Motif analyses were performed using SignalP (v5.0)[86] and TMHMM (v2.0)[87].

**Transmission electron microscopy**. All tissue processing, sectioning, and imaging was carried out at the Conventional Electron Microscopy Facility at Harvard Medical School. Forebrain tissues were fixed in 2.5% glutaraldehyde/2% paraformaldehyde in 0.1 M sodium cacodylate buffer (pH 7.4). They were then washed in 0.1 M cacodylate buffer and postfixed with 1% osmiumtetroxide (OsO4)/1.5% potassium ferrocyanide (KFeCN6) for 1 h, washed in water three times and incubated in 1% aqueous uranyl acetate for one hour. This was followed by two washes in water and subsequent dehydration in grades of alcohol (10 min each; 50%, 70%, 90%, 2 × 10 min 100%). Samples were then incubated in propyleneoxide for 1 h and infiltrated overnight in a 1:1 mixture of propyleneoxide and TAAB Epon (Marivac Canada Inc. St. Laurent, Canada). The following day, the samples were embedded in TAAB Epon and polymerized at 60 degrees C for 48 h. Ultrathin sections (about 80 nm) were cut on a Reichert Ultracut-S microtome, and picked up onto copper grids stained with lead citrate. Sections were examined in a JEOL 1200EX Transmission electron microscope or a TecnaiG² Spirit BioTWIN. Images were recorded with an AMT 2k CCD camera.

**Glycogen and mitochondrial quantification**. Glycogen and mitochondrial quantification was performed by hand using the ImageJ plugin FIJI[88,89]. Percentages were calculated by dividing the area of interest by the total area of ChP epithelial cell within the field of view. No other cell types were included in the analysis. For each condition, analyses were performed across multiple individual animals ($N = 3$ for each age). From each animal, 10–20 fields of view were imaged at 3000× for glycogen analysis and 5–10 fields of view were imaged at 3000× for mitochondrial analysis. Each different field of view represented a unique cell or cells, and fields of view were chosen such that both the apical and basal surfaces of the cell were visible. For mitochondrial distribution, a custom MatLab (v.2018) code was written to extract the centroid from mitochondria data traced in ImageJ ROIs (Supplementary file titled "Fig1_Supporting_MitoDistance.m"). Then a distance transformation was performed from each mitochondrion centroid to the hand-traced apical or basal surfaces. The shortest distance was extracted to calculate the apical: basal proximity ratio, such that 1 = on the apical surface and 0 = on the basal surface. The analyses included a total of 1747 adult mitochondria, 2241 P7 mitochondria, 2257 P0 mitochondria, and 1123 embryonic mitochondria. The same groups of embryonic and adult animals were used for the above analysis.

**Seahorse metabolic analysis**. ChP explants were dissected in HBSS (Fisher, SH30031FS) and maintained on wet ice until plated. Only the posterior leaflet of the P0, P7, and adult ChP was retained for analysis due to empirically determined limitations of the oxygen availability in the XFe96 Agilent Seahorse system. Tissue explants were plated on Seahorse XFe96 spheroid microplates (Agilent, 102905-100) coated with Cell TAK (Corning), in Seahorse XF Base Medium (Agilent, 102353-100) supplemented with 0.18% glucose, 1 mM L-glutamine, and 1 mM pyruvate at pH 7.4 and incubated for 1 h at 37 °C in a non-CO2 incubator. Extracellular acidification rates (ECAR) and oxygen consumption rates (OCR) were measured via the Cell Mito Stress Test (Agilent, 103015-100) with a Seahorse XFe96Analyzer (Agilent) following the manufacturer's protocols. Data were processed using Wave software (Agilent). ATP production was calculated as the difference in OCR measurements before and after oligomycin injection, as described by the manufacturer's protocol (Agilent, 103015-100). The Cell Mito Stress test was performed 2–5 independent times (5 for adult; 2 for P7; 2 for P0; 3 for E16.5). The individual analyses were performed by averaging the readings from both the right and left hemisphere lateral ventricle ChP for each individual. Data were normalized by Calcein-AM (2 μM in PBS, Life Technologies L-3224) fluorescence measured at the end of the assay. Each datum is the normalized average of the 2 LVChP from each individual normalized to the average of the adult levels run on the same plate for each assay to account for any experimental variability. Raw and processed files are included in source data.

**High $K^+$ challenge study**. Fresh LVChPs were dissected from P4 pups and adult mice in room temperature HBSS and glued down onto imaging dishes with coverslip bottom. The tissues were incubated at 37 °C with Calcein-AM (Invitrogen L-3224; 1:200) for 10 min and then rinsed with 37 °C artificial CSF (aCSF: 119 mM NaCl, 2.5 mM KCl, 26 mM NaHCO3, 1 mM NaH2PO4, 11 mM glucose, with fresh 1.0 mM magnesium chloride and 2.8 mM calcium chloride; for P4 experiments, 5 mM KCl was used to match with CSF [$K^+$]). Protein was not added to aCSF

because, while in rat, the protein concentration decreases 10× between P2 and adult[90]; this change is only 2× in mouse[45] and unlikely to contribute substantially to the driving force. The tissues were soaked in 1.8 ml aCSF at the beginning of each imaging session and allowed to stabilize for 10 min. One Z-stack was acquired to reflect the baseline cell volume. Then a 10× KCl solution in aCSF was spiked into the bath to bring the final bath K+ concentration to 50 mM immediately before imaging continued. A total of five 3D Z-stacks were acquired throughout a 10-min imaging session to capture changes in cellular volume over time. Each stack acquisition took less than 30 s to minimize changes in cell volume from the beginning to the end of each stack. All imaging studies were carried out at 37 °C. Image stacks were imported into Imaris (v7.7.1; Bitplane) software. Individual epithelial cells were identified by shape. Cells with discrete borders that were present at all timepoints and had dark pixels both above and below them in Z for the whole timecourse were selected a priori and then traced by hand using the "Surpass" functionality to create a 3D surface volume through all Z-stacks based on Calcein-AM uptake signal. Due to known z-step distance and interpolation between the planes, Imaris calculated the number of voxels for each cell. This analysis was then repeated for the same cell throughout the timecourse. We verified manually that the same cell was analyzed throughout the timecourse based on the topology of the surrounding cells, allowing for adjustment for any x–y drifting that occurred. The relative volume was calculated as $dV/V_0$ for each timepoint (t) where $V_0$ is the initial volume of the cell, t is each subsequent timepoint after addition of challenge, and $dV = Vt - V_0$. In LVChP from each animal, 5 distinct cells were analyzed and their values at each timepoint were averaged to represent the average value for each animal. A total of 4 animals ($N = 4$) were analyzed from each age group.

**Tissue processing**. Samples were fixed in 4% paraformaldehyde (PFA). For cryosectioning, samples were incubated in the following series of solutions: 10% sucrose, 20% sucrose, 30% sucrose, 1:1 mixture of 30% sucrose and OCT (overnight), and OCT (1 h). Samples were frozen in OCT.

**Immunostaining**. Cryosections were blocked and permeabilized (0.3% Triton X-100 in PBS; 5% serum), incubated in primary antibodies overnight and secondary antibodies for 2 h. Sections were counterstained with Hoechst 33342 (Invitrogen H3570, 1:10,000) and mounted using Fluoromount-G (SouthernBiotech). The following primary antibodies were used: chicken anti-GFP (Abcam ab13970; 1:1000), mouse anti-Aqp1 (Santa Cruz sc-32737; 1:100), rabbit anti-CHD4 (Abcam ab72418, 1:200), rabbit anti-NKCC1 (Abcam ab59791; 1:500), rat anti-HA (Roche 11867423001; 1:1000). Secondary antibodies were selected from the Alexa series (Invitrogen, 1:500). Images were acquired using Zeiss LSM880 confocal microscope with ×20 objective and ×63 oil objective. ZEN Black software was used for image acquisition and ZEN Blue used for Airy processing.

**Co-IP**. Tissues were homogenized in NET buffer (150 mM NaCl, 10 mM Tris 8.0, 5 mM EDTA, 10% glycerol and 2% Triton X-100) supplemented with protease inhibitors. Protein concentration was determined by BCA assay (Thermo Scientific 23227). Lysates with same amount of total protein (250–1000 µg based on experiments) were precleared at 4 °C for 2 hr with Protein G agarose and then incubated with desired antibody or control antibody at 4 °C overnight (no beads present during antibody incubation). Protein G agarose beads were added to lysate-antibody mixture after overnight incubation for 2 h. Beads were washed thoroughly and then eluted by boiling in 2% SDS. ChP tissues were pooled across seven litters of P0 pups and 30 adults to achieve sufficient protein for Co-IP.

**Immunoblotting**. Tissues were homogenized in RIPA buffer supplemented with protease and phosphatase inhibitors. Protein concentration was determined by BCA assay (Thermo Scientific 23227). Samples were denatured in 2% SDS with 2-mercaptoethanol by heating at 37 °C (for NKCC1) or 95 °C (for CHD4 and other NuRD complex proteins) for 5 min. Equal amounts of proteins were loaded and separated by electrophoresis in a 4–15% gradient polyacrylamide gel (BioRad #1653320) or NuPAGE 4–12% Bis-Tris gel (Invitrogen #NP0322), transferred to a nitrocellulose membrane (250 mA, 1.5 h, on ice), blocked in filtered 5% BSA or milk in TBST, incubated with primary antibodies overnight at 4 °C followed by HRP conjugated secondary antibodies (1:5000) for 1 h, and visualized with ECL substrate. For phosphorylated protein analysis, the phosphoproteins were probed first, and then blots were stripped (Thermo Scientific 21059) and reprobed for total proteins. For co-IP protein analysis, TrueBlot secondary antibody (eBioscience 18-8816-33) was used to detect only nondenatured IgG and avoid background signal from IP antibody. The following primary antibodies were used: rabbit anti-NKCC1 (Abcam ab59791; 1:1000), rabbit anti-pNKCC1 (EMD Millipore ABS1004; 1:1000), rabbit anti-ATP1a1 (Upstate C464.6/05-369; 1:250), goat anti-Klotho (R&D AF1819-sp; 1:200), rabbit anti-GAPDH (Sigma G9545; 1:10,000), mouse anti-HA (Abcam ab130275; 1:1000), rabbit anti-CHD4 (Abcam ab72418; 1:2000), rabbit anti-MBD3 (Abcam ab157464; 1:1000), rabbit anti-HDAC1 (Abcam ab7028; 1:2000), mouse anti-HDAC2 (Abcam 51832; 1:2000). The same protein samples were blotted for multiple targets to allow controlled quantitative comparison (i.e., NKCC1 and GAPDH). Raw blot images are included in Source Data.

**Quantitative RT-PCR**. For mRNA expression analyses, the ChP were dissected and pooled from 2 pups. RNA was isolated using the MirVana miRNA isolation kit (Invitrogen AM1561) following manufacturer's specifications without miRNA enrichment step. Extracted RNA was quantified spectrophotometrically and 100 ng was reverse-transcribed into cDNA using the High Capacity cDNA Reverse Transcription kit (Applied Biosystems #4368814) following manufacturer's specifications. RT-qPCRs were performed in duplicate using Taqman Gene Expression Assays and Taqman Gene Expression Master Mix (Applied Biosystems) with GAPDH as an internal control. Cycling was executed using the StepOnePlus Real-Time PCR System (Invitrogen) and analysis of relative gene expression was performed using the $2^{-\Delta\Delta CT}$ method. Technical replicates were averaged for their cycling thresholds and further calculations were performed with those means. Each RNA sample was analyzed for multiple genes.

**In utero intracerebroventricular injection (ICV)**. Timed pregnant mice (E14.5) were anesthetized with isoflurane and warmed. Laparotomy was performed. Embryos were stabilized by hand and lateral ventricles were visually identified. AAV solution was delivered into lateral ventricles using fine glass capillary pipettes with beveled tips. The AAV solution contained (0.01%) Fast Green FCF (Sigma F7252-5G) to aid with visualization. Uterine horns were returned into the abdominal space, and the incision was sutured. Meloxicam analgesia was longitudinally delivered according to IACUC protocol.

**Intraventricular kaolin injection in postnatal pups**. Postnatal day 4 pups (P4) were anesthetized. A small incision was made in the scalp, and 1 µl of sterile kaolin solution (25% in PBS) was injected into the left lateral ventricle using glass capillary pipettes. The injection site with corresponding lateral ventricle localization was determined as between bregma and lambda, and 1 mm from mid-line. The skin incision was sutured and reinforced with Vetbond (3M™ ID 70200742529). The pups were warmed and returned to the dam.

**AAV production**. The original AAV-NKCC1 plasmid was purchased from Addgene (pcDNA3.1 HA CFP hNKCC1 WT (NT15-H) was a gift from Biff Forbush: Addgene plasmid # 49077; http://n2t.net/addgene:49077; RRID: Addgene_49077). The plasmid carries an 3×HA tag at the N-terminal of NKCC1 to allow detection and separation from endogenous NKCC1. The CFP tag was removed by BsaI digestion to reduce insert size for AAV production. Virus production and purification were performed by the Penn Vector Core. The very large size of the plasmid resulted in variable transduction efficiency. All mice receiving AAV-NKCC1 were analyzed for HA expression after every experiment to confirm transduction efficiency. AAV-GFP and AAV-Cre were purchased from the Boston Children's Hospital Viral Core.

**Magnetic resonance imaging (MRI)**. Mice were imaged using Bruker BioSpec small animal MRI (7T) at 2 week and P50 while under anesthesia by isoflurane. A warm pad was used to maintain body temperature. Breathing rate and heart rate were monitored to reflect the depth of anesthesia. All axial T2 images were acquired using the following criteria: TE/TR = 60/4000; Ave = 8; RARE = 4; slice thickness = 0.6 mm. Ventricle volumes were calculated by manual segmentation using FIJI/ImageJ. Brain sizes were calculated by averaging four cross-sectional areas from the four highlighted slices (red) with visible LV and 3 V in control mice (corresponding slices were used in AAV-NKCC1 regardless of visibility of ventricles). Ventricle regions were not excluded from the brain size measurements. In studies with unilateral kaolin injection, 3D reconstruction of the ventricles was performed by manual segmentation in ITK-SNAP[91] and exported through ParaView.

**Constant rate CSF infusion test (ICP and compliance measurement)**. An apparatus was developed to perform a constant infusion test in mice through a single catheter for both infusion of CSF and monitoring of ICP. A 20-cc syringe was filled with aCSF and placed in an automated infusion pump (GenieTouch, Kent Scientific Co., Denver) set to deliver a constant rate infusion of 1–4 uL/min. The syringe was connected via pressure tubing to hemostasis valve Y connector (Qosina, NY). A fiberoptic ICP sensor (Fiso Technologies Inc, Quebec, Canada) was inserted through the other port of the rotating hemostat and then into 0.55 mm diameter catheter until the sensor was flush with the catheter's distal tip. The entire apparatus and tubing was carefully screened to ensure the absence of air bubbles. Adult mice were then deeply anesthetized, placed on a warm pad, and head-fixed with ear bars. The distal end of the infusion device (catheter with fiberoptic sensor) was placed inside the lateral ventricle through a hole made by a hand-held twist drill with a #74 wire gauge bit (−0.4 mm (anterior-posterior) and 1.2 mm (medial-lateral) with respect to Bregma, and a depth of 2 mm from the outer edge of the skull); the catheter was then sealed with Vetbond (3 M, Minnesota). Intraventricular access and water-tight seal was confirmed by observation of arterial and respiratory waveforms in the ICP signal and a transient rise in ICP upon gentle compression of the abdomen and thorax. Two min of baseline ICP were recorded before initiating the infusion of aCSF. As the infusion proceeded, careful observation was made of the mouse's respiratory rate. After the ICP level reached a new plateau, the infusion was discontinued. Parameters of the Marmarou

model of CSF dynamics for constant rate infusions were estimated by a non-linear least squares fit of the model to the ICP data[54]

$$ICP(t) = \frac{\left[i_{\text{infusion}} + \frac{ICP_{\text{baseline}} - p_0}{R_{\text{CSF}}}\right] \cdot [ICP_{\text{baseline}} - p_0]}{\frac{ICP_{\text{baseline}} - p_0}{R_{\text{CSF}}} + i_{\text{infusion}} \cdot \left[e^{-\left(\frac{i_{\text{infusion}} + \frac{ICP_{\text{baseline}} - p_0}{R_{\text{CSF}}}}{C_i}\right) \cdot t}\right]} \tag{1}$$

where $i_{\text{infusion}}$ is the rate of infusion, $ICP_{\text{baseline}}$ is the ICP level before infusion, $p_0$ is a pressure in the storage arm of the model, $R_{\text{CSF}}$ is the resistance to CSF outflow, and $C_i$ is the compliance coefficient.

**Statistics and reproducibility**. Biological replicates ($N$) were defined as samples from distinct and biologically independent individual animals, analyzed either in the same experiment or within multiple experiments, with the exception when individual animal could not provide sufficient sample (i.e., CSF), in which case multiple animals were pooled into one distinct biological replicate and most details are stated in the corresponding figure legends. Due to limited space, these metrics for Fig. 1 are listed here: Fig. 1a–b contain data from three independent experiments for $N = 6$ biological replicates. Figure 1c Data from two independent experiments with a total of three biologically independent samples. d–f. $N = 3$ distinct animals from two independent experiments (same images analyzed in d, e, and f), 5–10 field of view (FOV) per animal (See source data for exact numbers of FOV), distinct cells were captured in each FOV. Figure 1h, i $N = 26$ E16.5 embryos (4 litters); $N = 19$ P0 pups (2 litters); $N = 12$ P7 pups (2 litters); $N = 16$ adults. Data are from five independent experiments for adult; two independent experiments for P7 and P0; three independent experiments for E16.5. Each datapoint is an average of the 2 LVChP from an individual. j–k $N = 3$ animals per age; 433–833 mitochondria per individual.

The following measurements were taken using the same set of samples/animals: mitochondria/glycogen quantification, immunoblotting, qPCR, and different parameters from CSF infusion tests (ICP, compliance, $R_{\text{CSF}}$). In these cases the same image, protein or RNA samples, and mice were analyzed for multiple readouts and targets, to allow controlled quantitative comparison. Statistical analyses were performed using Prism 7 or R. Outliers were excluded using ROUT method ($Q = 1\%$). Appropriate statistical tests were selected based on the distribution of data, homogeneity of variances, and sample size. The majority of the analyses were done using one-way ANOVA with multiple comparison correction (Sidak) or Welch's two-tailed unpaired $t$-test, except for Fig. 1d–f, and Fig. 1h, i where the analysis was done by Welch's ANOVA with Dunnett's T3 multiple comparison test, and Fig. 1k–l where the analysis was done using Kolmogorov–Smirnov test. F tests or Bartlett's tests were used to assess homogeneity of variances between data sets. Parametric tests (t-test, ANOVA) were used only if data were normally distributed and variances were approximately equal. Otherwise, nonparametric alternatives were chosen. Data are presented as mean ± standard deviation (SD). If multiple measurements were taken from a single individual, data are presented as mean ± standard errors of the mean (SEMs). Please refer to figure legends for sample size and technical replicate numbers. $p$ values < 0.05 were considered significant ($*p < 0.05$, $**p < 0.01$, $***p < 0.001$, $****p < 0.0001$). Exact $p$ values can be found in the figure legends. $P$ values are also marked in the figures where space allows.

**Reporting summary**. Further information on research design is available in the Nature Research Reporting Summary linked to this article.

## Data availability
Sequencing data from TRAP study (for Fig. 2 and Supplementary Fig. 4) are available in GEO (https://www.ncbi.nlm.nih.gov/geo/query/acc.cgi?acc=GSE138970; accession number 138970). All other data are available from the authors. All biological material were either directly commercially available or are available upon request from the lab. Source data are provided with this paper.

## Code availability
The custom MatLab code is provided as supplementary for Fig. 1 and Supplementary Fig. 3 (Fig1_Supporting_MitoDistance.m). The code is also available on GitHub (https://github.com/LehtinenLab/Xu-Fame-2020). All other data are available in the main text or the supplementary materials.

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

## Acknowledgements

We thank members of the Lehtinen, Heiman, and Warf labs for helpful discussions; Nancy Chamberlin for advice on the manuscript; Katia Georgopoulos for sharing the *Chd4* fl/fl mouse line and associated genotyping methods; P. Ellen Grant for the ICP monitor. We thank the following facilities and personnel: Maria Ericsson and HMS EM facility; Yaotang Wu and Michael Marcotrigiano and BCH Small Animal Imaging Laboratory; the MIT BioMicro Center (TRAP sequencing); BCH viral core and University of Pennsylvania Vector Core; the IDDRC Cellular Imaging Core and Harvard Digestive Diseases Center Imaging Core. NIH T32 HL110852 (R.M.F. and J.C.); William Randolph Hearst Fund (J.C.); NSF Graduate Research Fellowship Program (F.B.S.); OFD/BTREC/CTREC Faculty Development Fellowship Award (R.M.F.); Simons Foundation Autism Research Awards (IDs 590293 and 645596 for C.N. and D.S., respectively). NIH R01 AI130591 and R35 HL145242 (M.J.H.); NIH R00 HD083512 (P.-Y.L.); and R01 HD096693 (P.-Y.L. and B.C.W.); BCH Pilot Grant, Pediatric Hydrocephalus Foundation, Hydrocephalus Association, Human Frontier Science Program (HFSP) research program grant #RGP0063/2018, NIH R01 NS088566, the New York Stem Cell Foundation (M.K.L.); BCH IDDRC 1U54HD090255 and BCH viral core P30EY012196. M.K.L. is a New York Stem Cell Foundation—Robertson Investigator.

## Author contributions

H.X., R.M.F., C.S., J.S., P.-Y.L., B.C.W., F.B.S., J.C., D.S., C.N., and M.K.L. designed and performed experiments; H.X., R.M.F., C.S., and J.S. analyzed the data; Y.Z. and M.J.H. provided material; A.V., F.G., and M.H. provided technological support; H.X., R.M.F., and M.K.L. wrote the manuscript. All co-authors edited the manuscript.

## Competing interests

The authors declare no competing interests.
