## [Peer Review File · Nature Communications]

Reviewers' Comments:

Reviewer #1:

Remarks to the Author:

This paper deals with the longstanding problem of the mechanism by which CSF is cleared from the brain in early stages of its development. This is a neglected area of research perhaps because the methods available to study it have been very limited.

There are two separate aspects of the problem. One is the route by which CSF leaves the brain early in development since in the very early stages of brain development after neural tube closure the developing brain grows around a closed ventricular system and only opens to the subarachnoid space with the onset of drainage at a later stage of development; this appears to occur around E17 in the rat fetus (Jones & Sellars, 1982; Johansson et al. 2008) with formation of the foramina of Luschka and Magendie at around the time of birth in the rat (Strong & Alban, 1932). The other aspect is the mechanism by which CSF is formed early in brain development. In this study the authors have interpreted their results as providing evidence that the choroid plexus in the newborn period in the mouse is a route by which CSF is drained from the brain, in advance of development of mechanisms that are characteristic of the adult brain.

The authors have provided a substantial amount of experimental detail which should be sufficient for anyone with to repeat parts of the study. The manuscripts and experimental details are presented clearly.

The primary observation of the study is that in the early postnatal period in the mouse there is a steep decline in K⁺ concentration in CSF. The authors cite a review for papers showing that CSF K⁺ concentration declines during development in a number of species. However, the only species with a gestational period and stage of brain development similar to the mouse is the rat. The data on that (Amtorp & Sørensen, 1974) show the CSF K⁺ concentration is unchanged from birth to postnatal (P) day 16 after which there is a slow decline towards the adult value. Values from Ferguson & Woodbury (1969) show a similar postnatal pattern. This is quite different from the steep decline between P0 and P7 identified in the mouse in Xu et al. This of course may represent a species difference, but does not justify the suggestion that this itself is a common feature across species. The paper presents serum K⁺ values at E14.5 and adult. Do the authors have values for serum K⁺ at the same critical postnatal ages as the CSF values? This would allow calculation of CSF/plasma ratios. An ion gradient higher than can be accounted by ultrafiltration is a cardinal sign for secretion of CSF (Davson, 1968). It is also evidence that tight junctions are sufficiently well developed to allow the gradient to exist. As pointed out by several of the authors cited in reference 15.

The study has focussed on the Na⁺-K⁺-Cl⁻ and water co-transporter, NKCC1, using molecular techniques which included ChP-targeted NKCC1 overexpression and NKCC1 OE mice. It would have been helpful for the uninitiated to include a definition of "OE". The rest of the study was concerned with intracranial pressure and volume changes in the developing and adult brain and the possible role of NKCC1. These experiments involved an ingenious adaptation of the Marmarou model combined with MRI imaging in adult rats to study how the brain and cranial spaces adapt to CSF volume changes. The authors also studied kaolin induced hydrocephalus at P4 and the effects of changing expression/activity of NKCC1.

The authors frequently refer to CSF clearance and their schematic in Figure 7 appears to suggest that the route is from the choroid plexus cells to the blood supply of the plexus and they suggest that this is a mechanism that may be specific to the developing brain occurring before other clearance mechanisms begin to function. However, they do not appear to provide any direct evidence for this. Efflux of high concentrations of K⁺ could have been studied using injection of ⁴²K into the ventricular system as has been done using ventriculo-cisternal perfusion in adult rabbits (Bradbury & Stulcova, 1970). These experiments showed that the adult brain has a substantial ability to remove K⁺ added to CSF; although these studies do not define the route, such experiments would provide direct evidence of efflux of K⁺ in the presence of raised CSF potassium in the developing brain as described by Bradbury & Stulcova (1970) in adult rabbits. Other experiments published by Bradbury and his colleagues have demonstrated a remarkable

ability of the brain barrier mechanisms to maintain a stable K⁺ concentration in CSF in the face of major changes in plasma K⁺ concentrations (e.g. Bradbury & Davson, 1965; Bradbury & Kleeman, 1967; Bradbury and Stulcova, 1970). The latter authors concluded that "The large increase in barrier clearance of 42K which occurs at levels of potassium in the perfusing fluid of 4-5 m-equiv/l. and above cannot be explained on any other basis than that there is a big increase in the passage of 42K ions across the choroid epithelium or across the endothelium of capillaries in brain tissue close to the ventricular system." This provides strong support for the conclusion of Xu et al that in the presence of high CSF potassium concentrations there is an outward efflux of K⁺ across the choroid plexus.

This control of CSF K⁺ concentration is an example of one of the most tightly controlled homeostatic mechanisms in the body. It is presumably essential for the normal functioning of the brain in terms of nerve impulse transmission, or as Hugh Davson has put it colloquially, without such control of the brain's internal environment K⁺ our sensory experience would be limited to a series of flashes and bangs. The findings of Xu et al and of earlier studies show that this K⁺ stability is also present in the developing brain and is presumably significant for aspects of its development.

In lines 71-73 the authors comment "Although the prevailing model posits that the ChP provides net unidirectional, luminal secretion of ions and water to form CSF, insufficient corroborating data have been collected under physiological experimental conditions. " The evidence is indeed sparse but there is some. For example, Amtorp & Sørensen (1974) showed that acetazolamide, an inhibitor of carbonic anhydrase had no effect on the CSF/plasma protein ratio at P0 but marginally increased it at P4, and substantially at P8 and P16 in rats. Carbonic anhydrase is a key component of the mechanism of CSF secretion by the choroid plexuses. The effect of acetazolamide suggests that the choroid plexus is contributing to CSF secretion early in neonatal life, which has been measured as early as P3 and increases by a factor of two by P8 (Bass & Lundborg, 1973; Johanson & Woodbury, 1974). Also, I do not think the authors take sufficient account of the substantial evidence that transport across the choroid plexuses is bidirectional, but there are quantitative differences in level of transfer that depend on the ions/molecules involved and local physiological or pathological conditions.

The authors mention several CSF clearance routes including arachnoid granulations, perineural and paravascular pathways, and meningeal lymphatics. The evidence for these drainage pathways is limited, even in the adult brain, but still less in development. The authors could perhaps consider the studies of Johnstone who mainly worked on lymphatic drainage in adult brain of several species but has published one short report on this route in neonatal and older rats (Johnston et al. 2006).

Overall, the authors have provided new data on the function of one of the many ion transport mechanisms in the developing choroid plexus and have shown that the plexus is involved in outward movement of K⁺ when it is present in CSF in excess concentrations as has been previously shown only in adult brain. They have thus provided evidence that an important homeostatic mechanism for control of CSF K⁺ is present early in brain development. But given that a similar mechanism appears to be present in adult choroid plexus it seems inappropriate to suggest that this mechanism is in some way unique to the developing brain.

Specific points.

The most appropriate units for electrolytes in body fluids when comparisons are being made are mequiv/Kg H₂O, as this allows for differences in protein concentration in the different fluids (Davson, 1968). This will be less critical in plasma/serum and CSF from fetal and neonatal animals as their CSF protein concentration is much higher and plasma protein much lower than in the adult (see reference 15).

Arachnoid granulations are only present in human meninges. In other mammalian species the equivalent tissue is arachnoid villi.

Line 306 "congenial fluid disorders". The authors presumably mean congenital.

Line 520 why did the authors use an artificial CSF with 2.5mM KCl for both postnatal and adult choroid plexus volume experiments when their results on K⁺ measurements show there is a substantial difference between the two ages? Also did the authors consider using an artificial CF

with a protein concentration appropriate for the ages of choroid plexus studied? In the rat the CSF protein concentration at P2 is nearly 10x that in the adult (Dziegielewska et al 1981).
Line 976. Neonatal (P0-7, left) is not left but above.

I thought this is a very nice piece of work, but perhaps the authors would be willing to have another look at their main conclusion.

Norman R Saunders
University of Melbourne
5 August 2020

References

- Amtorp O, Sørensen SC (1974). The ontogenetic development of concentration differences for protein and ions between plasma and cerebrospinal fluid in rabbits and rats. *J Physiol* 243, 387–400.
- Bass NH, Lundborg P (1973). Postnatal development of bulk flow in the cerebrospinal fluid system of the albino rat: clearance of carboxyl-(14C) inulin after intrathecal infusion. *Brain Res* 52, 323–332.
- Bradbury MW, Davson H. (1965). The transport of potassium between blood, cerebrospinal fluid and brain. *J Physiol*. 181(1), 151-174.
- Bradbury MW, Kleeman CR. (1967). Stability of the potassium content of cerebrospinal fluid and brain. *Am J Physiol*. 213(2), 519-28.
- Bradbury MW, Stulcová B. (1970). Efflux mechanism contributing to the stability of the potassium concentration in cerebrospinal fluid. *J Physiol*. 208(2), 415-430.
- Davson H (1967). *Physiology of the Cerebrospinal Fluid*, p35. Churchill, London.
- Dziegielewska KM et al. (1981). Proteins in cerebrospinal fluid and plasma of fetal rats during development. *Dev Biol* 83, 192–200.
- Ferguson RK, Woodbury DM (1969). Penetration of 14C-inulin and 14C-sucrose into brain, cerebrospinal fluid, and skeletal muscle of developing rats. *Exp Brain Res* 7, 181–194.
- Johanson CE, Woodbury DM (1974). Changes in CSF flow and extracellular space in the developing rat. In *Drugs and the Developing Brain*, ed. Vernadakis A & Weiner N, pp 281–287. Plenum Press, New York.
- Johansson PA et al. (2008). The blood-CSF barrier explained: when development is not immaturity. *Bioessays* 30, 237–248.
- Johnston M et al. (2006). Cerebrospinal fluid transport across the cribriform plate into extracranial lymphatics in rats: development and quantification. *Cerebrospinal Fluid Research* 2006, 3(Suppl 1):S9.
- Jones HC, Sellars RA. 1982. The movement of fluid out of the cerebral ventricles in fetal and neonatal rats. *Z Kinderchir* 37:130–133.
- Strong RM, Alban H. 1932. The development of the lateral apertures of the fourth ventricle in the albino rat. *Anat Rec* 52:39.

Reviewer #2:

Remarks to the Author:

In this intriguing study, the authors describe a mechanism whereby ions and water can be absorbed from CSF in the ventricle in developing mice via the NKCC1 transporter. The authors use a range of methods and present solid data in neonatal mice, however they should be careful in their extrapolation of these data to adult mice or adult pathologies.

Major:

1) Role of the choroid plexus in development vs. adult

The authors study changes in the composition of CSF in the postnatal period until adulthood is reached, however detailed investigation and interventions (e.g. AAV, kaolin) are focused on the

earlier time points. From the data shown, the authors cannot claim that this mechanism of CSF absorption by the choroid plexus occurs in adulthood, or that they have evidence that this could be a target in adult pathologies (line 319). They have also not shown that there is no CSF secretion by the choroid plexus during the "transitional developmental phase" (line 333), only that there is absorption of water and K⁺ in the choroid plexus epithelial cells via NKCC1.

The changes observed, e.g. in mitochondrial localisation, which temporally coincide with changes in CSF K⁺ concentration, are likely a part of choroid plexus maturation, rather than occurring primarily to fuel K⁺ absorption, since the localisation is maintained after the K⁺ clearance phase has ended.

Can the authors suggest why the differences in ventricle size after NKCC1 overexpression were maintained until P50? Are the authors suggesting that even as adulthood is reached, NKCC1 may not be secreting CSF (e.g. Figure 7)? The authors use terms like "the transporter's native directionality" (line 302) but also state that an ion gradient powers this transporter (line 161). Thus, it seems like the directionality of the NKCC1 transporter may depend on physiological parameters, and not one fixed "native directionality" throughout development.

2) Other outflow routes – use of "canonical" vs "non-canonical": The terms canonical and non-canonical do not add to the clarity of the manuscript. Particularly the description of other recently described (and controversial) CSF outflow routes as "canonical", e.g. dural lymphatics, is highly debatable. The authors state that other outflow routes are not yet fully developed and therefore "not available to contribute to CSF dynamics" (line 63). However, the extent of the contribution of different outflow routes during development is not established. Solid physiological and anatomical work from the group of Miles Johnston indicates that the CSF outflow routes through the cribriform plate and along other cranial nerves are present either before or after birth depending on the species (Papaiconomou et al, 2002, 10.1152/ajpregu.00173.2002; Koh et al, 2006; 10.1007/s00429-006-0085-1). In mouse, unlike dural lymphatics, lymphatics outside of the skull develop before birth and may contribute to outflow of CSF along the exiting cranial nerves. In the text (e.g. line 281, 376), the authors should consistently mention the full list of alternative outflow routes mentioned in the introduction (line 60) and not make conclusive statements that other CSF outflow sites are not yet developed.

3) Lack of citations to earlier work: There is extensive work on the development of the choroid plexus that is not discussed. A consensus was reached that the choroid plexus starts to secrete fluid shortly after birth. Additionally, research in rats has already demonstrated that CSF/plasma ratios of potassium start to decline at around the time of birth (Ferguson and Woodbury, 1969; *Exp Brain Res* 7:181–194). Also, CSF outflow resistance has been examined in newborn rats (Deane and Jones, *Z Kinderchir* 38:64). The findings of the current project should be put into context with this earlier research and further justification for the new hypothesis should be given. One might start with the discussion in the aforementioned Koh et al, 2006 reference (10.1007/s00429-006-0085-1) as it examines the earlier research from a similar angle of developing clearance pathways.

4) Title - The authors should choose a title which better reflects their data. The current title does not mention that the choroid plexus is involved, or that this mechanism was studied in development. The use of the term "non-canonical" creates some confusion and a clearer term to indicate that this mechanism is not yet widely recognized should be chosen.

5) The CSF K⁺ concentration shows a peak at P0. Why were Na⁺ and Cl⁻ only assessed at E14.5 and adult? Does K⁺ show a significant difference between the time points assessed for these ions? If NKCC1 is a co-transporter of all 3 ions, and is playing a fundamental role, why are Na⁺ and Cl⁻ concentrations not altered? Are these being transported out of the CSF via other transporters?

6) Aqp1 – This water channel is included in some analysis and also in the schematic in Fig. 2e, but its putative contribution (or lack thereof?) to the proposed mechanism is not discussed, in fact Aqp1 is only mentioned in the antibodies section.

Minor:

-line 36 "more permissive cerebral hydrodynamics" - this is very vague. Please clarify what exactly.

-Line 57, 306: congenial -> congenital

- “OE” -> write out “overexpression”
- expand method section – in utero LV injection – how is the embryo held in place? How is the ventricle located? ...
- use of “2mo” vs “adult” in text. This could be defined once, and then a consistent term should be used
- line 136 “unbiasedly” - rephrase
- line 147 “specifically” – this term suggest that other pathways are not enriched, which is not the case
- line 155 “specifically” – this term suggests that other organs are not gaining fluid and ion modulatory functions postnatally
- line 227 – if more water is being removed, wouldn’t the authors expect the protein concentration to increase?
- line 237 “final measurement at P50” – the methods indicate that measurements were made at P14 and P50 only. The phrasing here suggests that there were more time points in between.
- line 255 – C_i is inversely proportional to the rate of ICP increase
- line 300 – add “at specific developmental timepoint” or similar
- line 318 – add references
- line 334 – and where do the ions come from?
- line 336 -> “which have been reported to secrete CSF”
- line 535 – is a cell an individual?
- Figure 5a-c: schematic of brain slices on left. Were all of the shown slices (black) used to calculate brain cross sectional area? The text states all slices with a ventricle (LV and 3V and 4V? clarify please) visible but the shown slices seem to include other areas. Is the ventricular area subtracted from the brain cross sectional area on each section? If not, if the ventricle size is decreasing (“deflated”), shouldn’t the measured brain size also decrease?
- Supplementary table 1, line 296 – The authors wish to indicate in this table that the measurement of intracellular ion concentrations in choroid plexus epithelial cells gives a large range of values, however they should make it clearer that the included concentrations come from a number of species and age points.
- Supplementary figure 3b – suggestion to reformat the presentation of the upper/lower quartiles, as the current presentations makes the “dashed line” look like selected data points
- Supplementary figure 4g – How fixed is the localisation of these transporters? Is it possible that they may change their localisation during maturation of the CP?
- Supplementary figure 5a – match y-axis scale to Fig. 2f.
- Supplementary figure 5b – inclusion of pNKCC1?

Response to Reviewers

REVIEWER COMMENTS (*responses are in Italic*)

Reviewer #1 (Remarks to the Author):

This paper deals with the longstanding problem of the mechanism by which CSF is cleared from the brain in early stages of its development. This is a neglected area of research perhaps because the methods available to study it have been very limited. There are two separate aspects of the problem. One is the route by which CSF leaves the brain early in development since in the very early stages of brain development after neural tube closure the developing brain grows around a closed ventricular system and only opens to the subarachnoid space with the onset of drainage at a later stage of development; this appears to occur around E17 in the rat fetus (Jones & Sellars, 1982; Johansson et al. 2008) with formation of the foramina of Luschka and Magendie at around the time of birth in the rat (Strong & Alban, 1932). The other aspect is the mechanism by which CSF is formed early in brain development. In this study the authors have interpreted their results as providing evidence that the choroid plexus in the newborn period in the mouse is a route by which CSF is drained from the brain, in advance of development of mechanisms that are characteristic of the adult brain.

The authors have provided a substantial amount of experimental detail which should be sufficient for anyone with to repeat parts of the study. The manuscripts and experimental details are presented clearly.

We thank the reviewer for this summary and positive feedback on this study.

1. The primary observation of the study is that in the early postnatal period in the mouse there is a steep decline in K⁺ concentration in CSF. The authors cite a review for papers showing that CSF K⁺ concentration declines during development in a number of species. However, the only species with a gestational period and stage of brain development similar to the mouse is the rat. The data on that (Amtorp & Sørensen, 1974) show the CSF K⁺ concentration is unchanged from birth to postnatal (P) day 16 after which there is a slow decline towards the adult value. Values from Ferguson & Woodbury (1969) show a similar postnatal pattern. This is quite different from the steep decline between P0 and P7 identified in the mouse in Xu et al. This of course may represent a species difference, but does not justify the suggestion that this itself is a common feature across species.

We thank the reviewer for bringing up this good point. As suggested, we have changed the wording to state that the CSF K⁺ decrease is conserved, although the magnitude and timing differs by species (line109-112).

-The paper presents serum K⁺ values at E14.5 and adult. Do the authors have values for serum K⁺ at the same critical postnatal ages as the CSF values? This would

allow calculation of CSF/plasma ratios. An ion gradient higher than can be accounted by ultrafiltration is a cardinal sign for secretion of CSF (Davson, 1968). It is also evidence that tight junctions are sufficiently well developed to allow the gradient to exist. As pointed out by several of the authors cited in reference 15.

Thank you for this suggestion. We measured serum [K⁺] at additional time points and calculated CSF/plasma ratios (Fig. 1b. The data are also summarized in Table R1 below). We also reference Davson 1968 here (line 109).

Table R1. Serum [K⁺] and CSF/serum [K⁺] ratio at various developmental ages

	Serum [K⁺] (mM)	CSF/serum [K⁺] ratio
E14.5	15.45 ± 0.21	0.59 ± 0.14
P0	10.52 ± 4.59	0.91 ± 0.33
P7	11.71 ± 3.07	0.37 ± 0.08
P14	7.41 ± 1.93	0.44 ± 0.02
Adult	9.11 ± 1.41	0.41 ± 0.08

2. The study has focused on the Na⁺-K⁺-Cl⁻ and water co-transporter, NKCC1, using molecular techniques which included ChP-targeted NKCC1 overexpression and NKCC1 OE mice.

-It would have been helpful for the uninitiated to include a definition of “OE”. The rest of the study was concerned with intracranial pressure and volume changes in the developing and adult brain and the possible role of NKCC1. These experiments involved an ingenious adaptation of the Marmarou model combined with MRI imaging in adult rats to study how the brain and cranial spaces adapt to CSF volume changes. The authors also studied kaolin induced hydrocephalus at P4 and the effects of changing expression/activity of NKCC1.

We now define OE as “overexpression” in line 235.

We are pleased that the reviewer finds our adaptation of the Marmarou model in mice particularly compelling.

3. The authors frequently refer to CSF clearance and their schematic in Figure 7 appears to suggest that the route is from the choroid plexus cells to the blood supply of the plexus and they suggest that this is a mechanism that may be specific to the developing brain occurring before other clearance mechanisms begin to function. However, they do not appear to provide any direct evidence for this. Efflux of high concentrations of K⁺ could have been studied using injection of ⁴²K into the ventricular system as has been done using ventriculo-cisternal perfusion in adult rabbits (Bradbury & Stulcova, 1970). These experiments showed that the adult brain has a substantial ability to remove K⁺ added to CSF; although these studies do not define the route, such experiments would provide direct evidence of efflux of K⁺ in the presence of raised CSF potassium in the developing brain as described by Bradbury & Stulcova (1970) in adult

rabbits. Other experiments published by Bradbury and his colleagues have demonstrated a remarkable ability of the brain barrier mechanisms to maintain a stable K⁺ concentration in CSF in the face of major changes in plasma K⁺ concentrations (e.g. Bradbury & Davson, 1965; Bradbury & Kleeman, 1967; Bradbury and Stulcova, 1970). The latter authors concluded that “The large increase in barrier clearance of 42K which occurs at levels of potassium in the perfusing fluid of 4-5 m-equiv/l. and above cannot be explained on any other basis than that there is a big increase in the passage of 42K ions across the choroid epithelium or across the endothelium of capillaries in brain tissue close to the ventricular system.” This provides strong support for the conclusion of Xu et al that in the presence of high CSF potassium concentrations there is an outward efflux of K⁺ across the choroid plexus.

This control of CSF K⁺ concentration is an example of one of the most tightly controlled homeostatic mechanisms in the body. It is presumably essential for the normal functioning of the brain in terms of nerve impulse transmission, or as Hugh Davson has put it colloquially, without such control of the brain’s internal environment K⁺ our sensory experience would be limited to a series of flashes and bangs. The findings of Xu et al and of earlier studies show that this K⁺ stability is also present in the developing brain and is presumably significant for aspects of its development.

We agree with the reviewer’s remarks which indicate a deep appreciation for the importance and complexity of K⁺ regulation in the nervous system. We agree that we did not directly demonstrate the route for K⁺ movement from the ChP to the blood and that this is a hypothesis rather than a conclusion. We have amended our Discussion accordingly (line 325). We also discuss how published work, including the experiments performed by Bradbury & Stulcova, 1970 in adult rats, is consistent with our hypothesis (lines 112-115) (Bradbury & Stulcova, 1970; Bradbury & Davson, 1965; Bradbury & Kleeman, 1967). Although perfusion and detection of radioactive isotopes in mouse pups is experimentally impossible at this time, we agree that these studies would be helpful.

4. In lines 71-73 the authors comment “Although the prevailing model posits that the ChP provides net unidirectional, luminal secretion of ions and water to form CSF, insufficient corroborating data have been collected under physiological experimental conditions. ” The evidence is indeed sparse but there is some. For example, Amtorp & Sørensen (1974) showed that acetazolamide, an inhibitor of carbonic anhydrase inhibitor had no effect on the CSF/plasma protein ratio at P0 but marginally increased it at P4, and substantially at P8 and P16 in rats. Carbonic anhydrase is a key component of the mechanism of CSF secretion by the choroid plexuses. The effect of acetazolamide suggests that the choroid plexus is contributing to CSF secretion early in neonatal life, which has been measured as early as P3 and increases by a factor of two by P8 (Bass & Lundborg, 1973; Johanson & Woodbury, 1974).

- Also, I do not think the authors take sufficient account of the substantial evidence that transport across the choroid plexuses is bidirectional, but there are

quantitative differences in level of transfer that depend on the ions/molecules involved and local physiological or pathological conditions.

We thank the reviewer for providing this context which we have now incorporated into our text. We now include the suggested references (Amtorp & Sørensen (1974); Bass & Lundborg, 1973; Johanson & Woodbury, 1974) to emphasize the sparse, but important evidence of ChP luminal secretion (line 79). Additionally, we place more emphasis on ChP-to-CSF secretory direction in previous works to highlight in contrast to the special developmental phase when CSF-to-ChP transport level exceeds the other. We also include additional discussion of the substantial studies supporting the bidirectionality of ChP transport, and clarify that we do not claim identification of a new trafficking direction, but rather we highlight the net clearance effect of NKCC1 which has, until now, been less discussed than its net secreting effects in the context of the ChP (lines 81-82, 91-93, 321-324).

5. The authors mention several CSF clearance routes including arachnoid granulations, perineural and paravascular pathways, and meningeal lymphatics. The evidence for these drainage pathways is limited, even in the adult brain, but still less in development.

- The authors could perhaps consider the studies of Johnstone who mainly worked on lymphatic drainage in adult brain of several species but has published one short report on this route in neonatal and older rats (Johnston et al. 2006).

We agree evidence for these drainage pathways and their developmental time-courses are limited. We thank the reviewer for pointing out Johnston's studies. We have now included more details about the current knowledge in regard to CSF drainage pathways and their development and cited the literature in our introduction (line 58-74).

6. Overall, the authors have provided new data on the function of one of the many ion transport mechanisms in the developing choroid plexus and have shown that the plexus is involved in outward movement of K^+ when it is present in CSF in excess concentrations as has been previously shown only in adult brain. They have thus provided evidence that an important homeostatic mechanism for control of CSF K^+ is present early in brain development. But given that a similar mechanism appears to be present in adult choroid plexus it seems inappropriate to suggest that this mechanism is in some way unique to the developing brain.

We thank the reviewer for accurately summarizing our findings. We did not intend to imply that this mechanism is unique to the developing brain. In fact, we believe it may come into play in any circumstance of elevated CSF $[K^+]$.

The difference between adult and postnatal cases is that while outward trafficking from CSF through the ChP was reported in adults when CSF has higher than homeostatic $[K^+]$ (Bradbury & Stulcova, 1970), in postnatal mice the mechanism occurs without challenge. We now clarify

that these mechanisms can occur at either early postnatal or adult stages and included relevant references to avoid future confusion (line 112-115, and 356-358).

Specific points.

- **The most appropriate units for electrolytes in body fluids when comparisons are being made are mequiv/Kg H₂O, as this allows for differences in protein concentration in the different fluids (Davson, 1968). This will be less critical in plasma/serum and CSF from fetal and neonatal animals as their CSF protein concentration is much higher and plasma protein much lower than in the adult (see reference 15).**

We thank the reviewer for pointing out the appropriate units. We chose to use mM to report the absolute concentration of K⁺ in a given fluid to avoid calculation artifacts. It is technically very difficult in embryonic and postnatal mouse samples to calculate the accurate Kg H₂O the way Davson, et al. did, mostly due to very limited sample volume. As the reviewer commented, the unit is less critical in this case of fetal and neonatal serum and CSF study. Therefore we believe that using mM minimizes potential calculation errors and anticipate that the unit is more familiar to readers who don't specialize in fluid biology.

- **Arachnoid granulations are only present in human meninges. In other mammalian species the equivalent tissue is arachnoid villi.**

Thank you, we correct this throughout.

- **Line 306 “congenial fluid disorders”. The authors presumably mean congenital.**

We have now corrected this typographical error.

- **Line 520 why did the authors use an artificial CSF with 2.5mM KCl for both postnatal and adult choroid plexus volume experiments when their results on K⁺ measurements show there is a substantial difference between the two ages? Also did the authors consider using an artificial CF with a protein concentration appropriate for the ages of choroid plexus studied? In the rat the CSF protein concentration at P2 is nearly 10x that in the adult (Dziegielewska et al 1981).**

Thank you for these questions. We chose not to include protein in the aCSF to be consistent with the more widely used aCSF composition as reported in the literature. Moreover, in mice CSF protein concentration changes no more than 2mg/ml during postnatal development (Lun, et al., J. Neurosci. 2015 Fig. 6G), and therefore contributes minimally to overall osmolarity. We have provided this rationale in the revised methods section (line 570-572) citing Dziegielewska et al., 1981 and Lun et al., 2015.

For similar reasons we initially used standard ion concentrations in our aCSF. But we have come to the same conclusion as the reviewer, that it would be best to use age-appropriate ion

concentrations in aCSF. Therefore, we repeated some experiments using aCSF that matches the ionic concentrations of the P4 natural CSF, including Na^+ , K^+ , Cl^- , Ca^{2+} , and Mg^{2+} (Fig. 2, Fig. S6, and updated methods).

In addition, our revised results include ion concentrations in CSF at several key developmental stages in mice, including embryonic (E14.5), neonatal (P0 and P4), and later postnatal (P7 and P14) and we discuss the potential impact of age-appropriate aCSF formulation, especially in regard to K^+ (line 410-415) to help inform future developmental brain study designs that incorporate aCSF (Supplementary Table 1).

- **Line 976. Neonatal (P0-7, left) is not left but above.**

We have now corrected this typographical error.

I thought this is a very nice piece of work, but perhaps the authors would be willing to have another look at their main conclusion.

We thank Prof Saunders for the careful evaluation of our work and have now included more detailed clarification throughout the manuscript to refine the conclusion: while there is bi-directionality of transport through ChP at all ages, we emphasize that early postnatal CSF has high K^+ that drives CSF-to-ChP transport through ChP NKCC1 which has the effect of net clearance of CSF.

Norman R Saunders
University of Melbourne
5 August 2020

References

1. Amtorp O, Sørensen SC (1974). The ontogenetic development of concentration differences for protein and ions between plasma and cerebrospinal fluid in rabbits and rats. *J Physiol* 243, 387–400.
2. Bass NH, Lundborg P (1973). Postnatal development of bulk flow in the cerebrospinal fluid system of the albino rat: clearance of carboxyl-(14C) inulin after intrathecal infusion. *Brain Res* 52, 323–332.
3. Bradbury MW, Davson H. (1965). The transport of potassium between blood, cerebrospinal fluid and brain. *J Physiol.* 181(1), 151-174.
4. Bradbury MW, Kleeman CR. (1967). Stability of the potassium content of cerebrospinal fluid and brain. *Am J Physiol.* 213(2), 519-28.
5. Bradbury MW, Stulcová B. (1970). Efflux mechanism contributing to the stability of the potassium concentration in cerebrospinal fluid. *J Physiol.* 208(2), 415-430.
6. Davson H (1967). *Physiology of the Cerebrospinal Fluid*, p35. Churchill, London.
7. Dziegielewska KM et al. (1981). Proteins in cerebrospinal fluid and plasma of fetal rats during development. *Dev Biol* 83, 192–200.

8. Ferguson RK, Woodbury DM (1969). Penetration of ¹⁴C-inulin and ¹⁴C-sucrose into brain, cerebrospinal fluid, and skeletal muscle of developing rats. *Exp Brain Res* 7, 181–194.
9. Johanson CE, Woodbury DM (1974). Changes in CSF flow and extracellular space in the developing rat. In *Drugs and the Developing Brain*, ed. Vernadakis A & Weiner N, pp 281–287. Plenum Press, New York.
10. Johansson PA et al. (2008). The blood-CSF barrier explained: when development is not immaturity. *Bioessays* 30, 237–248.
11. Johnston M et al. (2006). Cerebrospinal fluid transport across the cribriform plate into extracranial lymphatics in rats: development and quantification. *Cerebrospinal Fluid Research* 2006, 3(Suppl 1):S9.
12. Jones HC, Sellars RA. 1982. The movement of fluid out of the cerebral ventricles in fetal and neonatal rats. *Z Kinderchir* 37:130–133.
13. Strong RM, Alban H. 1932. The development of the lateral apertures of the fourth ventricle in the albino rat. *Anat Rec* 52:39.

Thank you for these extremely important suggestions. We have included most of these references in the revised manuscript as noted in the point-by-point response above.

Reviewer #2 (Remarks to the Author):

In this intriguing study, the authors describe a mechanism whereby ions and water can be absorbed from CSF in the ventricle in developing mice via the NKCC1 transporter. The authors use a range of methods and present solid data in neonatal mice, however they should be careful in their extrapolation of these data to adult mice or adult pathologies.

We thank the reviewer for this summary and positive feedback on this study. We address individual points below.

Major:

1) Role of the choroid plexus in development vs. adult

- **The authors study changes in the composition of CSF in the postnatal period until adulthood is reached, however detailed investigation and interventions (e.g. AAV, kaolin) are focused on the earlier time points. From the data shown, the authors cannot claim that this mechanism of CSF absorption by the choroid plexus occurs in adulthood, or that they have evidence that this could be a target in adult pathologies (line 319). They have also not shown that there is no CSF secretion by the choroid plexus during the “transitional developmental phase” (line 333), only that there is absorption of water and K⁺ in the choroid plexus epithelial cells via NKCC1.**

This point is well-taken as Reviewer 1 also raised concerns regarding our clarity in relating immature to adult animals. In this revision we have attempted to carefully distinguish our claims vs speculations and contextualize them with regard to the NKCC1 mechanism per se vs. the

overall net secretory or absorptive role of the ChP under different physiological and developmental conditions. The reviewer is correct that in this particular study we present no evidence that this mechanism occurs in adulthood. Our logic that it might be is based on the presumption that the directionality of K^+ and water transport via NKCC1 is determined by the relevant ion gradients and that these gradients could theoretically favor CSF absorption under some circumstances in adults.

We now include references of CSF K^+ clearance in an adult rodent study. We clarify that we speculate, based on our data and previous studies, that similar clearance mechanisms could occur in adult brains (line 356-358). In line 372-375, we now use more precise wording to state that ChP epithelial cells mediate CSF water and K^+ absorption via NKCC1, but we make no claim of whether ChP is producing CSF or not during the period discussed.

- **The changes observed, e.g. in mitochondrial localisation, which temporally coincide with changes in CSF K^+ concentration, are likely a part of choroid plexus maturation, rather than occurring primarily to fuel K^+ absorption, since the localisation is maintained after the K^+ clearance phase has ended.**

We agree with the reviewer and have now clarified this point (line 147-150). We now state that the overall maturation of the ChP involves metabolic upregulation that, among other processes, is consistent with supporting increased ion transport demands.

- **Can the authors suggest why the differences in ventricle size after NKCC1 overexpression were maintained until P50? Are the authors suggesting that even as adulthood is reached, NKCC1 may not be secreting CSF (e.g. Figure 7)? The authors use terms like “the transporter’s native directionality” (line 302) but also state that an ion gradient powers this transporter (line 161). Thus, it seems like the directionality of the NKCC1 transporter may depend on physiological parameters, and not one fixed “native directionality” throughout development.**

Indeed, “native directionality” was a confusing choice of words and we have altered our text accordingly. The short answer to the ventricle size question is that we do not know. However, we now include more discussion regarding the P50 ventricle size data (line 262-267). Although we would not expect enlargement as a result of OE in adult mice because phosphorylation, rather than protein levels, is rate-limiting at this age, we cannot explain the lasting reduction in size.

We suspect the sustained differences in ventricle sizes may reflect a mechanical effect on properties of the ventricular ependymal cell layer that outlasts the stimulus (low pressure from ventricle collapse). It would be interesting to study the cellular and elastic properties of this tissue in the future.

2) Other outflow routes – use of “canonical” vs “non-canonical”: The terms canonical and non-canonical do not add to the clarity of the manuscript. Particularly the description of other recently described (and controversial) CSF outflow routes as “canonical”, e.g. dural lymphatics, is highly debatable. The authors state that other outflow routes are not yet fully developed and therefore “not available to contribute to CSF dynamics” (line 63). However, the extent of the contribution of different outflow routes during development is not established. Solid physiological and anatomical work from the group of Miles Johnston indicates that the CSF outflow routes through the cribriform plate and along other cranial nerves are present either before or after birth depending on the species (Papaiconomou et al, 2002, 10.1152/ajpregu.00173.2002; Koh et al, 2006; 10.1007/s00429-006-0085-1). In mouse, unlike dural lymphatics, lymphatics outside of the skull develop before birth and may contribute to outflow of CSF along the exiting cranial nerves. In the text (e.g. line 281, 376), the authors should consistently mention the full list of alternative outflow routes mentioned in the introduction (line 60) and not make conclusive statements that other CSF outflow sites are not yet developed.

In this revision we consistently list all of the known clearance routes (e.g. line 59-61 and line 315-316) and have added more details about the current knowledge of other outflow routes in the introduction, including additional references provided by the reviewer. We clarified that the known clearance routes still have open questions, such as those around the lymphatics systems. Instead of implying that other routes are not well-established, we included the relevant literature and discussed what is known about the development of CSF drainage routes and pointed out the developmental time-courses of the proposed drainage routes are still not fully established. We replaced the term “canonical” with “known” or “reported”.

3) Lack of citations to earlier work: There is extensive work on the development of the choroid plexus that is not discussed. A consensus was reached that the choroid plexus starts to secrete fluid shortly after birth. Additionally, research in rats has already demonstrated that CSF/plasma ratios of potassium start to decline at around the time of birth (Ferguson and Woodbury, 1969; Exp Brain Res 7:181–194). Also, CSF outflow resistance has been examined in newborn rats (Deane and Jones, Z Kinderchir 38:64). The findings of the current project should be put into context with this earlier research and further justification for the new hypothesis should be given. One might start with the discussion in the aforementioned Koh et al, 2006 reference (10.1007/s00429-006-0085-1) as it examines the earlier research from a similar angle of developing clearance pathways.

We thank the reviewer for suggesting additional relevant scholarship. These references have now been added as key components of our discussion on pages 3-4, 6, and 13-14.

4) Title - The authors should choose a title which better reflects their data. The current title does not mention that the choroid plexus is involved, or that this mechanism was studied in development. The use of the term “non-canonical” creates some confusion

and a clearer term to indicate that this mechanism is not yet widely recognized should be chosen.

We have revised our title to reflect the reviewer's comments. The manuscript is now titled "Choroid plexus NKCC1 mediates perinatal cerebrospinal fluid clearance".

5) The CSF K⁺ concentration shows a peak at P0.

- Why were Na⁺ and Cl⁻ only assessed at E14.5 and adult?
- Does K⁺ show a significant difference between the time points assessed for these ions?
- If NKCC1 is a co-transporter of all 3 ions, and is playing a fundamental role, why are Na⁺ and Cl⁻ concentrations not altered?
- Are these being transported out of the CSF via other transporters?

*To provide a more comprehensive view, we added levels of all three relevant ions, Na⁺, K⁺, and Cl⁻, for each timepoint from E14.5 to adult. The statistical comparisons of all ions were performed between a developing age and adult. In summary, Na⁺ remains minimally changed across ages, Cl⁻ showed small but statistically significant increase over time, while K⁺ decreased significantly during the first postnatal week. We now include all the data in **Supplementary Table 1** and adjusted the text accordingly.*

While NKCC1 is a co-transporter of all 3 ions, there are other transporters and channels that can transport each of the three ions across the ChP-CSF border, including but not limited to KCC4 (K-Cl co-transporter 2), NHE1 (Na/H exchanger 1), NBCe2 (Na/Bicarbonate cotransporter 2), and several volume-regulated Cl⁻ channels. Because all the transporters work at different rates and respond to different ion gradients, it is not expected that all three ions change their concentrations in a synchronized manner through NKCC1 activity alone. It is likely that Na⁺ and Cl⁻ concentrations are maintained as they are by multiple transporters.

6) Aqp1 – This water channel is included in some analysis and also in the schematic in Fig. 2e, but its putative contribution (or lack thereof?) to the proposed mechanism is not discussed, in fact Aqp1 is only mentioned in the antibodies section.

We now include Aqp1 in the results (line 173-175). In summary, Aqp1 did not show significant change in transcription or translation during development, unlike other candidates (NKCC1, ATP1a1, ATP1b1, KL). AQP1 is a water channel that is driven by osmolarity instead of specific ions' gradients, and therefore Aqp1 is not part of the proposed mechanism involving changing ions.

Minor:

-line 36 "more permissive cerebral hydrodynamics" - this is very vague. Please clarify what exactly.

We were attempting to describe in plain language an increase in the cerebral compliance which would flatten the relationship between a change in CSF volume and the resultant change in intracranial pressure. Initially we avoided the term compliance because the concept of compliance may be unfamiliar to readers not specialized in brain fluid biology. We have now clarified this concept.

-Line 57, 306: congenial -> congenital

We have now corrected this typographical error.

-“OE” -> write out “overexpression”

We have now updated this section to include the full definition before the abbreviation (line 235).

-expand method section – in utero LV injection – how is the embryo held in place? How is the ventricle located? ...

We have now included more method details for in utero LV injection (line 648-655).

-use of “2mo” vs “adult” in text. This could be defined once, and then a consistent term should be used

We have now changed all to “adult” after it is defined.

-line 136 “unbiasedly” - rephrase

We have now removed the word “unbiasedly” and rephrase the sentences (line 155). We intended to point out the screening approach did not have any researcher selection bias towards any given transporters.

-line 147 “specifically” – this term suggest that other pathways are not enriched, which is not the case

We were hoping to emphasize cation transport pathways among all transmembrane pathways. We have removed the word to avoid confusion and changed it to “notably” (line 166).

-line 155 “specifically” – this term suggests that other organs are not gaining fluid and ion modulatory functions postnatally

We have removed the word to avoid confusion. We wanted to emphasize that ChP gained these functions postnatally, and did not intend to compare with other organs.

-line 227 – if more water is being removed, wouldn't the authors expect the protein concentration to increase?

Not necessarily. The protein concentration in the CSF can be modulated by many mechanisms, including but not limited to, CSF water secretion and removal, protein secretion and uptake by the ChP and other surrounding cells, and overall exchange with interstitial fluid. Therefore, it is not a given that the amount of water removed through ChP NKCC1 would cause a sustained increase in protein concentration. Nonetheless, there may have been a transient increase of protein concentration that we did not detect, which was subsequently normalized by other mechanisms modulating CSF protein concentration, but our data do not speak to this.

-line 237 “final measurement at P50” – the methods indicate that measurements were made at P14 and P50 only. The phrasing here suggests that there were more time points in between.

We changed the wording to be more clear (line 262).

-line 255 – Ci is inversely proportional to the rate of ICP increase

We changed the text accordingly (lines 289-290).

-line 300 – add “at specific developmental timepoint” or similar

We added “from a specific developmental stage onward”, as the AAV is introduced at E14.5 and continues to express the gene of interest afterwards (line 339-340).

-line 318 – add references

We included references and rephrased to avoid misunderstanding between speculation and conclusion (now line 322-326).

-line 334 – and where do the ions come from?

We thank the reviewer for pointing out this omission. We added “ions” to the sentences in question (line 374).

-line 336 -> “which have been reported to secrete CSF”

We modified the text accordingly (lines 377-378).

-line 535 – is a cell an individual?

We added more details and rephrased to clarify the analysis process. Each individual animal has 5 cells analyzed, and a total of 4 individual animals were analyzed for each experimental group (lines 590-593).

-Figure 5a-c: schematic of brain slices on left. Were all of the shown slices (black) used to calculate brain cross sectional area? The text states all slices with a ventricle (LV and 3V and 4V? clarify please) visible but the shown slices seem to include other areas. Is the ventricular area subtracted from the brain cross sectional area on each section? If not, if the ventricle size is decreasing (“deflated”), shouldn’t the measured brain size also decrease?

The black slices indicate all the slices we imaged in each mouse. The red ones were highlighted as those with visible LV and 3V space in uninjected control mice, and shown in the accompanying panel. We used only the red slices to calculate brain cross-sectional area. In NKCC1 OE mice, because the ventricles were smaller, only a subset of the red slices had visible ventricles, but all red slices were presented regardless of ventricle view to compare with control mice. We hope this answers the reviewer’s question about “other areas”. We have edited the methods (line 678-681) and figure legends to clarify.

In the data presented, ventricular area was not subtracted from the brain cross-sectional area. However, because the ventricle area only represents ~1% of the brain cross-sectional area, while the relative standard deviation in brain sizes within each experimental group is around 4%, we do not expect any detectable decrease in brain size due to ventricle shrinkage alone. We have performed the analysis with ventricles subtracted and the results are below for the reviewer. The methods have now been updated to be explicit about ventricle inclusion (line 681).

Fig. R1. Average brain size comparison between AAV-GFP mice and AAV-NKCC1 mice. The ventricles were excluded from the quantification.

-Supplementary table 1, line 296 – The authors wish to indicate in this table that the measurement of intracellular ion concentrations in choroid plexus epithelial cells gives a large range of values, however they should make it clearer that the included concentrations come from a number of species and age points.

Additional columns have now been included for “species” and “developmental stage”.

-Supplementary figure 3b – suggestion to reformat the presentation of the upper/lower quartiles, as the current presentations makes the “dashed line” look like selected data points

We have modified the figure as suggested to improve the presentation.

-Supplementary figure 4g – How fixed is the localisation of these transporters? Is it possible that they may change their localisation during maturation of the CP?

We now include new figure panels (Fig.S7) showing ChP NKCC1 consistently localizes to the apical surface of ChP epithelial cells.

-Supplementary figure 5a – match y-axis scale to Fig. 2f.

Thank you, this has been corrected.

-Supplementary figure 5b – inclusion of pNKCC1?

We now include the pNKCC1 panel in Fig. S5b.

Reviewers' Comments:

Reviewer #1:

Remarks to the Author:

I thank the authors for their comprehensive responses to my comments. I congratulate them on a significant contribution to this field

Reviewer #2:

Remarks to the Author:

In their revision of the manuscript, the authors have carefully responded to our comments and the majority of our concerns have been addressed. However, a few points remain, see below.

Previous Major comment 1) – development vs. adult

Line 93: "influenced the development of CSF ion and fluid homeostasis" – The choice of the word "development" is a bit confusing, can homeostasis develop? is it not defined as being at a state of stability? Suggestion to change to "influenced the establishment of CSF ion and fluid homeostasis"

Lines 320-322 – The sentence is also not clear. Is the time of development also a time of homeostasis? Suggestion to rephrase this whole sentence.

Previous Major comment 2) – other outflow routes

Line 58, line 315, line 421: While the authors have replaced the use of the word "canonical" when describing the outflow routes, the wording here is still too strong and suggests that all of the routes listed are firmly established. As some of these routes remain a matter of controversy, the authors should replace words like "known" with "putative" or "suggested", or otherwise rephrase the relevant sentences to make this clear.

Line 68: this is the first mention of the "glymphatic" concept. This reviewer suggests that this not be mentioned prominently as a CSF outflow route, since this is far from established. Rather, it is a putative route of CSF influx into the brain or its paravascular spaces, and thus far less relevant for the present study. Similarly, in line 423, "glymphatics" should perhaps not be included in a listing of putative CSF outflow routes. Other reported outflow routes (perineural, arachnoid villi/granulations) should instead be included in the list.

Line 316 – The authors should again cite the studies listed in introduction, not only the developmental studies which are cited here. This should also include references to Johnston papers cited elsewhere (12,13).

Previous minor comments:

Line 57: Please also correct the instance of the typographical error in line 57 (congenial -> congenital)

Additional comments:

Line 432: "previously unidentified route". In the view of this reviewer, the authors are primarily reporting a "previously undescribed phase" rather than "previously unidentified route" of clearance. Alternatively, the authors could simply remove the phrase "previously unidentified", with the sentence retaining its meaning and power.

Line 431 "presents" -> "suggests", "implicates"

Line 432 "known" -> "described", "reported", "putative"

Line 478: "QC" – define abbreviation.

Figure S7 – Are the authors trying to show that the transporter is not expressed on the basal side? If so, the location of the basal membrane should be somehow identified (staining or labelling) on the images. Perhaps higher magnification images could aid in this?

Response to Reviewers:

REVIEWERS' COMMENTS (*responses are in Italic*)

Reviewer #1 (Remarks to the Author):

I thank the authors for their comprehensive responses to my comments. I congratulate them on a significant contribution to this field

We thank Dr. Saunders for constructive comments and support!

Reviewer #2 (Remarks to the Author):

In their revision of the manuscript, the authors have carefully responded to our comments and the majority of our concerns have been addressed. However, a few points remain, see below.

Previous Major comment 1) – development vs. adult

Line 93: “influenced the development of CSF ion and fluid homeostasis” – The choice of the word “development” is a bit confusing, can homeostasis develop? is it not defined as being at a state of stability? Suggestion to change to “influenced the establishment of CSF ion and fluid homeostasis”

We thank the Reviewer for the suggestion to improve clarity. We have made the edits accordingly.

Lines 320-322 – The sentence is also not clear. Is the time of development also a time of homeostasis? Suggestion to rephrase this whole sentence.

We have changed the sentence to “This CSF clearance by the ChP during normal development contrasts the prevailing view that ChP always has net secretion of CSF at all times.” for improved clarity.

Previous Major comment 2) – other outflow routes

Line 58, line 315, line 421: While the authors have replaced the use of the word “canonical” when describing the outflow routes, the wording here is still too strong and suggests that all of the routes listed are firmly established. As some of these routes remain a matter of controversy, the authors should replace words like “known” with “putative” or “suggested”, or otherwise rephrase the relevant sentences to make this clear.

We have made these changes accordingly, and used “putative” and “suggested” where needed. We thank the Reviewer for providing more accurate word choices.

Line 68: this is the first mention of the “glymphatic” concept. This reviewer suggests that this not be mentioned prominently as a CSF outflow route, since this is far from established. Rather, it is a putative

route of CSF influx into the brain or its paravascular spaces, and thus far less relevant for the present study. Similarly, in line 423, “glymphatics” should perhaps not be included in a listing of putative CSF outflow routes. Other reported outflow routes (perineural, arachnoid villi/granulations) should instead be included in the list.

We have removed mentions of glymphatics from the manuscript.

Line 316 – The authors should again cite the studies listed in introduction, not only the developmental studies which are cited here. This should also include references to Johnston papers cited elsewhere (12,13).

We have included all the suggested citations.

Previous minor comments:

Line 57: Please also correct the instance of the typographical error in line 57 (congenial -> congenital)

We thank the reviewer for finding this typographical error. It is now corrected.

Additional comments:

Line 432: “previously unidentified route”. In the view of this reviewer, the authors are primarily reporting a “previously undescribed phase” rather than “previously unidentified route” of clearance. Alternatively, the authors could simply remove the phrase “previously unidentified”, with the sentence retaining its meaning and power.

We have removed the phrase “previously unidentified”.

Line 431 “presents” -> “suggests”, “implicates”

Line 432 “known” -> “described”, “reported”, “putative”

We have made these changes accordingly: “presents” → “implicates”, and “known” → “described”

Line 478: “QC” – define abbreviation.

We have added the definition (QC = quality control).

Figure S7 – Are the authors trying to show that the transporter is not expressed on the basal side? If so, the location of the basal membrane should be somehow identified (staining or labelling) on the images. Perhaps higher magnification images could aid in this?

We thank the reviewer for this advice. The transporter is indeed not expressed on the basal side, and we were hoping to deliver this message by showing co-localization of a validated apical membrane marker, AQP1. We now added labeling on the images to demonstrate the basal side of the epithelial layer in the merged image.